# Adaptive Self-improvement LLM Agentic System for ML Library Development

**Genghan Zhang** [1]  **Weixin Liang** [1]  **Olivia Hsu** [1]  **Kunle Olukotun** [1]

## Abstract

ML libraries, often written in architecture-specific programming languages (ASPLs) that target domain-specific architectures, are key to efficient ML systems. However, writing these high-performance ML libraries is challenging because it requires expert knowledge of both ML algorithms and the ASPL. Large language models (LLMs), on the other hand, have shown general coding capabilities. However, challenges remain when using LLMs for generating ML libraries using ASPLs because 1) this task is complicated even for human experts and 2) there are limited code examples due to the esoteric and evolving nature of ASPLs. We present an adaptive self-improvement agentic system that enables LLMs to perform such complex reasoning under limited data by iteratively improving their capability through self-generated experience. In order to evaluate the effectiveness of our system, we construct a benchmark of a typical ML library and generate ASPL code with both open and closed-source LLMs on this benchmark. Our results show improvements of up to $3.9\times$ over a baseline single LLM [1].

## 1. Introduction

With the ending of Dennard Scaling and Moore's Law, computer architectures are specializing in domain applications to achieve greater performance and efficiency and will continue to do so (Hennessy & Patterson, 2019). New domain-specific architectures (DSA) typically come with new architecture-specific programming languages (ASPL), such as CUDA for NVIDIA GPUs (NVIDIA, 2025), HIP for AMD GPUs (AMD, 2025), and Pallas for Google TPUs (Google, 2025). Even existing ASPLs change as

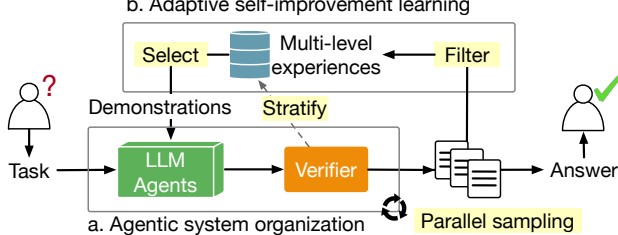

*Figure 1.* We propose an adaptive self-improvement LLM agentic system. LLM agents start from their base knowledge and accumulate experiences through parallel sampling. Our adaptive self-improvement learning algorithm filters high-quality answers, stratifies the earned experiences by difficulty, and adaptively selects demonstrations to enhance LLM agents.

generations of DSAs evolve because new DSAs introduce specialized functions to these existing ASPLs (Choquette, 2023). Efficiently utilizing these new functions requires fundamentally different programming styles and thus new ASPLs (Thakkar et al., 2023; Hagedorn et al., 2023).

Each DSA needs a corresponding ML library, a collection of ML operators written in its ASPL, before programmers can effectively use the DSA to accelerate ML applications. ML library development is challenging as it requires expertise in both ML algorithms and the target ASPL. Essentially, library development is a generation process that composes low-level ASPL primitives into high-level ML operators (Dong et al., 2024; Ye et al., 2025).

Furthermore, ML library development using ASPLs requires complex reasoning while minimizing data requirements. ML libraries development often occurs in parallel with hardware manufacturing to meet production deadlines (Villa et al., 2021). This time-constrained library–chip co-design process leaves limited code examples. Moreover, this task is complicated even for expert human programmers. For example, the publication of FlashAttention-3 (Shah et al., 2024) lagged behind the release of the H100 by two years. The challenge is further intensified by the need for ML libraries to co-evolve with new hardware to sustain performance. For instance, directly adapting FlashAttention-2 (Dao, 2023) from the A100 to H100 GPU witnessed a 47% performance drop (Spector et al., 2024).

[1]Department of Computer Science, Stanford University, USA. Correspondence to: Genghan Zhang <zgh23@stanford.edu>.

*Proceedings of the $42^{nd}$ International Conference on Machine Learning*, Vancouver, Canada. PMLR 267, 2025. Copyright 2025 by the author(s).

[1]The example code is public at https://github.com/zhang677/PCL-lite

The challenges in ML library development call for more automatic solutions. Furthermore, these automatic solutions need to self-improve to perform complex reasoning starting from simple and limited data. Large language models (LLMs) have demonstrated emerging capabilities in code generation (Kaplan et al., 2020; Wei et al., 2022a). Moreover, empirical evidence implies that LLMs already have the base knowledge of ML algorithms (Ouyang et al., 2024). Therefore, we explore the use of LLM agents to develop ML libraries with emerging ASPLs.

Current self-improvement methods for LLM agents fall short because of limited exploration or low data efficiency. LLM agents can enhance their performance by synthesizing semantically similar data (Yu et al., 2023; Shinn et al., 2024; Zhao et al., 2024). Although these methods are effective for local exploration (Chen et al., 2024), they are insufficient for tasks that require substantial cognitive effort (Huang et al., 2023). Self-improvement learning can significantly improve reasoning ability through reinforcement learning (Cobbe et al., 2021; Bai et al., 2022; Singh et al., 2023). This approach, however, currently requires hundreds of effective trajectories sampled from LLM agents for each problem (Wang et al., 2024), making it unsuitable for complex scenarios with limited data availability.

To address these limitations, we design an adaptive self-improvement learning algorithm integrated with an agentic system organization. This approach not only produces a self-improving agentic system to assist humans but also generates high-quality ML operators that can be leveraged by other systems. We show our system in Figure 1. Similar to human experiential learning (Kolb, 2014), our techniques create a self-improvement cycle: LLM agents evolve through earned experiences and these evolved agents can earn more experiences. This self-improvement cycle is fully automated, involving no human experts beyond the ASPL designers themselves, who initially tell the models how to use the ASPL primitives.

Our algorithm prioritizes hard-earned experiences gained from completing challenging tasks, inspired by curriculum learning (Bengio et al., 2009). When these hard-earned experiences are exhausted, the algorithm adaptively increases the number of demonstrations by incoporating experiences from less challenging tasks. Section 6.1 shows that hard-earned experiences improve LLM agents more efficiently than mixed ones. While mixed experiences may dilute demonstrations, they can help agents overcome learning obstacles and complete more tasks. As a byproduct, the algorithm adaptively increases test-time compute on challenging tasks until they are complete or the data is exhausted.

To emulate the library–chip co-design process, we choose Streaming Tensor Programs (STeP) as our target ASPL for library generation. STeP is an emerging ASPL designed for next-generation reconfigurable dataflow architectures (Prabhakar et al., 2017), a family of DSAs for AI (Prabhakar & Jairath, 2021; Chen et al., 2023; Prabhakar et al., 2024). The only public document of STeP is a non-archival three-page workshop publication (Sohn et al., 2024a), which defines its semantics without any code examples or execution environments. Therefore, STeP programs do not exist in the training corpus of any LLM.

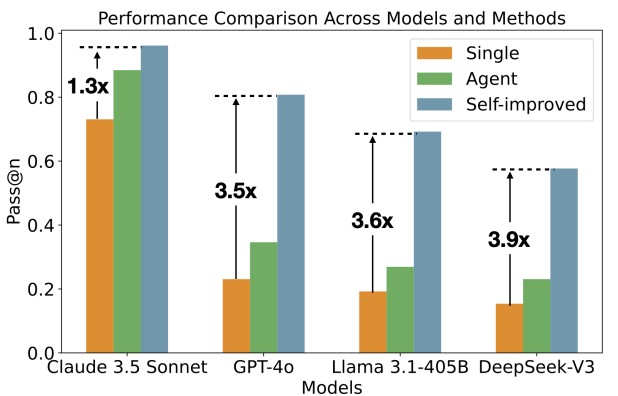

*Figure 2.* Portion of completed tasks (Pass@n) across models using single LLM, agentic system, and adaptive self-improvement agentic system, highlighting performance improvement.

Putting all these together, our system solves up to 96% of the tasks in our benchmark and achieves up to a $3.9\times$ improvement over a baseline single LLM, as shown in Figure 2. The contributions of this paper are: (1) an adaptive self-improvement learning algorithm that enables LLM agents to continuously construct ML libraries through adaptive experience-driven evolution; (2) an end-to-end agentic system that uses adaptive self-improvement to develop an ML library for STeP, an ASPL for a next-generation AI accelerator; (3) a complete evaluation of the adaptive self-improvement learning algorithm and the integrated agentic system on a realistic benchmark constructed from common ML operators.

## 2. Background

In this section, we provide background on how DSAs are programmed through their ASPLs, describe how these ASPLs are used to create end-user ML libraries, and identify key challenges of this library generation process. We also establish STeP as the target ASPL to explore LLM techniques for ML library development. Key concepts related to the background are listed in Table 8 for reference.

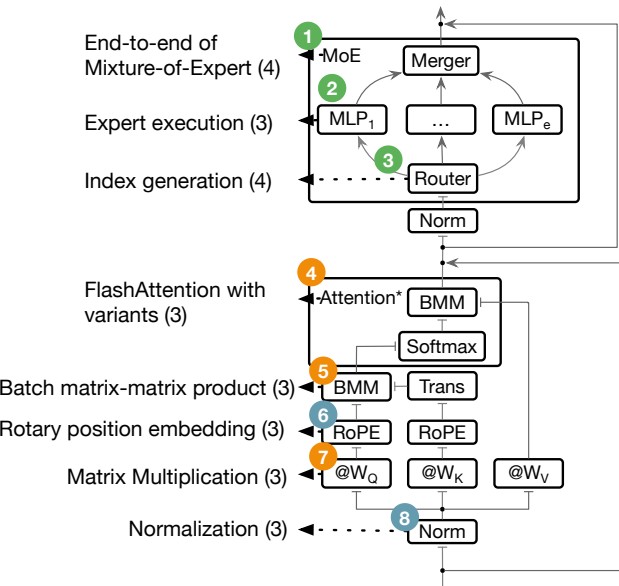

*Figure 3.* We benchmarked eight groups of ML operators within a common LLM layer (Jiang et al., 2024). These operators are categorized as dynamic, static matrix, and static vector, with parentheses indicating the number of tasks associated with each group.

## 2.1. Architecture-specific programming languages

ASPLs describe the low-level programming interface of a DSA using primitives and specialized functions. Primitives model the basic execution pattern similar to general-purpose programming language constructs, and specialized functions control specialized accelerator units that are optimized for domain applications on the DSA. Unlike domain-specific languages (DSLs), which are a top-down distillation of the domain algorithms, ASPLs refer to a bottom-up abstraction of the underlying chip architecture.

## 2.2. ML library development using ASPLs

ML libraries developed in ASPLs face portability challenges because ASPLs rapidly evolve to align with DSA updates in new generations to meet the demands of growing ML workloads. For example, the matrix multiplication units on NVIDIA GPUs and their corresponding MMA instructions have been updated every generation of GPU since their introduction (NVIDIA, 2017). Consequently, every library function that uses MMA instructions must be rewritten in a new ASPL (e.g. CUDA with new instructions) per generation. Moreover, ML libraries need to be shipped at the same time as the chip. In this case, library development costs are no longer negligible.

To solve these challenges, we propose to enhance users' learning capabilities for a given ASPL. This approach offers

an alternative to current automation techniques that focus on simplifying the learning curve for new ASPLs. Mainstream automation techniques compromise by optimizing ML operators whose performance is most significantly affected by the ASPL update (Tillet et al., 2019; Thakkar et al., 2023). Only focusing on certain ML operators does not fully incorporate some ASPL updates, such as memory optimizations, which could potentially accelerate any ML operator. Meanwhile, new ML operators are being proposed (Gu & Dao, 2023; Sun et al., 2024). Given these factors, we need better automation to improve the productivity of ML library development using APSLs.

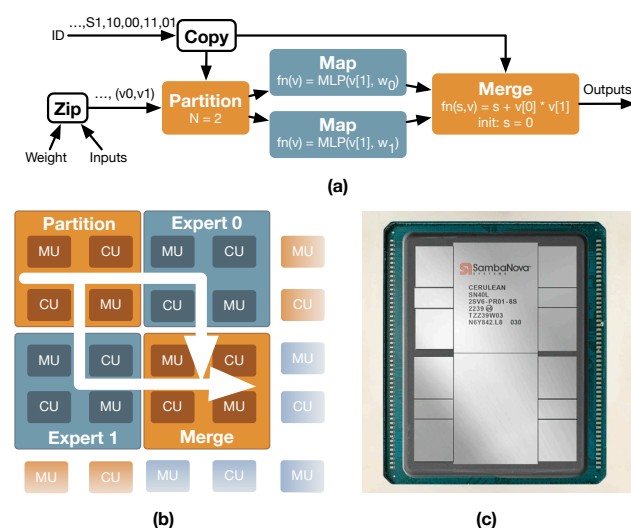

*Figure 4.* STeP is an ASPL for next-generation RDAs that models streaming dataflow execution. (a) STeP example program for a simplified MoE module. More details can be found in Appendix A.1. (b) An illustration of the streaming dataflow execution of (a) on an RDA of memory units (MU), and compute units (CU). (c) The SN40L, a deployed RDA chip.

## 2.3. STeP for next-generation RDA

We chose STeP as our target ASPL due to its potential for better efficiency and status as a research prototype ASPL. STeP's efficiency potential stems from its role as an ASPL for next-generation RDAs, which are a promising alternative to GPUs. The SN40L, a deployed RDA implementation shown in Figure 4(c), demonstrates record-breaking inference speeds for the Llama 3.1 405B model (SambaNova, 2024). Although STeP does not yet have a path to a fabricated chip, developing ML libraries in STeP still presents similar challenges as other ASPLs. Writing STeP programs requires complex reasoning about streaming dataflow execution, and our work began without any existing executable STeP programs to reference.

Similar to other ASPLs, primitives and specialized functions

compose to form operational semantics in STeP. Specifically, STeP primitives describe different stream token manipulation strategies, and STeP specialized functions express different configurations of RDA units. Inspired by parallel patterns and array programming (Hsu et al., 2023; Rucker et al., 2024), the streaming dataflow execution model in STeP treats data as streams of tokens flowing between computational units in space. STeP extends the conventional dataflow execution model of RDAs by introducing streaming semantics, as shown in Figure 4(b). This approach unifies data values and control signals into stream tokens, embedding control flow directly into the data to enable greater dynamism. A stream can be consumed by at most one primitive because of the queueing nature of dataflow (Zhang et al., 2021), which is called an affine type constraint in programming language theory (Wikipedia, 2024). The affine type constraint is a global property of the program since it counts the usage of a variable in the whole program.

STeP primitives are categorized as either arithmetic or shape manipulation. Arithmetic primitives apply computations and control flow to stream tokens. Shape manipulation primitives reshape the data within the stream by changing the control tokens. Figure 4(a) and Figure 17 are example STeP programs that contain 5 arithmetic primitives and only shape manipulation primitives, respectively.

---

**Algorithm 1** Adaptive self-improvement learning

**input** $\mathcal{X}$: task set, $m$: adaptive granularity
**Require:** $\theta$: LLM agentic system, $r$: reward from verifier, $\sigma$: filter function, $\beta$: selection function
1: $\mathcal{D} \leftarrow \emptyset$
2: $t \leftarrow 0$           $\triangleright$ iteration
3: **repeat**
4:   $\mathcal{E} \leftarrow \beta(\mathcal{D}, m)$       $\triangleright$ stratification
5:   **for** $e_j \in \mathcal{E}$ **do**
6:    $d_j \leftarrow [e_0, e_1, ... e_j]$      $\triangleright$ selection
7:    // Parallel sampling
8:    $\mathcal{C}_t \leftarrow \{\mathbb{E}_{y \sim p_\theta(y|x_i, d_j)}[r(x_i, y)] \mid x_i \in \mathcal{X}\}$
9:    $\mathcal{B}_t \leftarrow \{(x_i, y) \mid r(x_i, y) = 1, x_i \in \mathcal{X}\}$
10:    $\mathcal{S}_t \leftarrow \{x_i \mid c_i > 0, c_i \in \mathcal{C}_t\}$
11:    **if** $\mathcal{B}_t \neq \emptyset$ **then**
12:     $\mathcal{D} \leftarrow \mathcal{D} \cup \sigma(\mathcal{B}_t, \mathcal{C}_t, \mathcal{S}_t)$    $\triangleright$ filtering
13:     $\mathcal{X} \leftarrow \mathcal{X} \setminus \mathcal{S}_t$
14:     $t \leftarrow t + 1$
15:     **break**
16:    **end if**
17:    $t \leftarrow t + 1$
18:   **end for**
19: **until** $\mathcal{X} = \emptyset \vee (d_j = \mathcal{D} \wedge \mathcal{B}_{t-1} = \emptyset)$
**output** Solutions: $\mathcal{D}$

---

# 3. Adaptive self-improvement learning

Our adaptive self-improvement learning evolves LLM agentic systems with data generated by the systems themselves. This algorithm samples the agentic system in parallel for correct answers with success rates, filters high-quality correct answers, stratifies the earned experiences, and adaptively updates demonstrations until all the tasks are solved or the demonstrations are exhausted. The complete algorithm for adaptive self-improvement learning is shown in Algorithm 1. As a byproduct, the algorithm adaptively assigns more test-time compute to harder tasks by excluding tasks from the task set after completion. Figure 5 illustrates a running example in which only the unfinished tasks are fed to the agentic system. Additionally, this algorithm is independent of the specific organization of the agentic system, as shown in Figure 1.

## 3.1. Filtering high-quality answers

The filter function, $\sigma$, collects earned experience $\mathcal{D}$ by filtering one answer for each newly solved task in $\mathcal{S}_t$ and records the success rate of this answer in $\mathcal{C}_t$. $\sigma$ first groups the correct answers $\mathcal{B}_t$ by the isomorphic abstract syntax tree (Knuth, 1968). Then, $\sigma$ groups equivalent answers using AST isomorphism and randomly selects one representative from each group. Among these representatives, $\sigma$ chooses the one with the minimal length of pure code (excluding comments and empty lines) as the final answer for the task in $\mathcal{S}_t$. $\sigma$ also stores the success rate from $\mathcal{C}_t$ for $\beta$. The minimal length selection follows the Minimum Description Length principle for higher information density (Rissanen, 1978). On the other hand, shorter text might lose chain-of-thought comments (Wei et al., 2022b). We purposefully introduce randomness to the selection of representative answers for each isomorphic group to balance these two contradicting intuitions.

## 3.2. Stratification and selection

The selection function $\beta$ stratifies the earned experiences $\mathcal{D}$ by binning them into $m$ levels of difficulty and demonstrations $d_j$ are selected incrementally from the stratified experiences $\mathcal{E}$. We define the difficulty as the opposite of the success rate following (Lightman et al., 2023). $\beta$ sorts $\mathcal{D}$ in ascending order of success rate and then bins the tasks as evenly as possible to get the boundaries of each bin. Then tasks are rebinned using these boundary values. This selection strategy can cause repetitive steps as exemplified by the dashed-line circles (Iter 4&5 in Figure 7(a)) when the newly finished tasks are easier than the demonstrations. Our algorithm keeps these repetitive steps instead of avoiding them because the tasks with low success rates have a better chance of getting one correct answer with the number of samples doubled. For example, Iter 4 performs better than

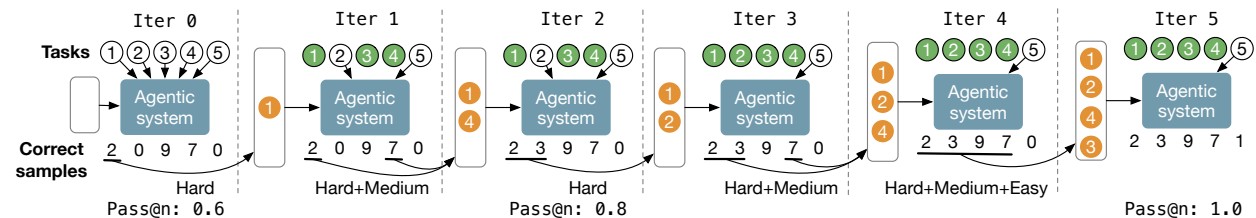

Figure 5. Running example of Algorithm 1 with $m = 3$ and $|\mathcal{X}| = 5$. Orange circles are demonstrations $d_j$ for the current iteration. Green circles are finished tasks and white ones are not. $m = 3$ means $\beta$ stratifies demonstrations into 3 levels: hard, medium, and easy. Iter 1 and 3 consider hard-only examples, and the other iterations consider mixed examples.

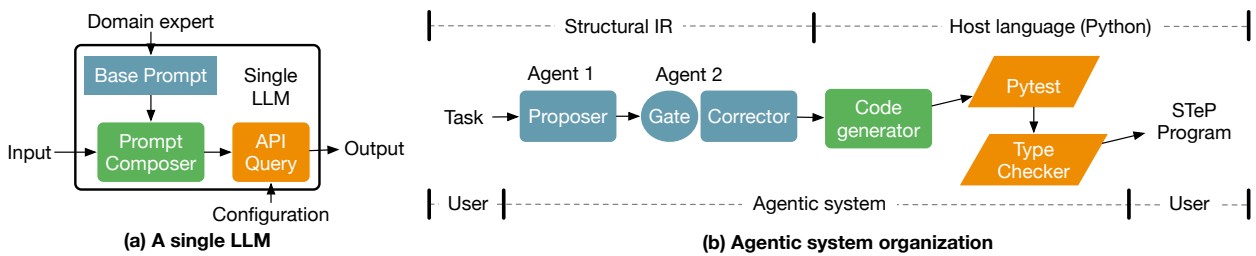

Figure 6. Details of our agentic system organization. (a) shows a single LLM. (b) shows the system components and their representations. The user is either a human or a self-improvement learning process. The filled colors align with the text colors in Section 4.

Iter 3 in Figure 7(b) with the same demonstrations. This method can also cause later iterations to have fewer tokens but with higher quality than previous iterations (Iter 2&3 in Figure 7(a)) when the boundary value crosses two bins and the newly finished tasks are easier.

### 3.3. Discussion

Algorithm 1 can also extend to new tasks that typically involve new ML operators and new hardware specialized functions. These can be incorporated into the initial task pool $\mathcal{X}$ and handled using the same adaptive self-improvement learning process by selectively sampling only the new tasks. Each example takes about 500 tokens, so cutting-edge LLMs can handle hundreds more tasks. If the number exceeds the context length, better stratification and selection functions are needed to preserve experience quality within the context window limit. If the task set $\mathcal{X}$ is finite, as in our case, the algorithm will terminate.

## 4. Agentic system organization

In this section, we introduce a specific agentic system organization tailored for ML library development using STeP as shown in Figure 6. The agentic system comprises LLM agents, a code generator, and verifiers, with a structural intermediate representation as the interface between users and system components.

### 4.1. LLM agents

As shown in Figure 6(a), each single LLM is designed for a specific purpose assigned by the domain expert through the base prompt. The base prompt contains the task description, base knowledge, and demonstrations. The prompt composer chains the base prompt and task-specific input, which is then fed into a configured LLM serving API.

Specifically, we design two agents, *a proposer and a guardian*. As explained in Section 2.3, the affine type constraint in STeP is a global property that requires thinking back and forth beyond step-by-step reasoning, which is challenging for the causal generation of LLMs. Therefore, we design a guardian agent to check and correct the affine type error globally. We exclude the demonstration tasks in the base prompt of the agents from the benchmark to avoid the model directly copying the answer.

The proposer agent generates a candidate STeP implementation whose base prompt comes from ASPL designers. The base prompt is composed of STeP references and usage patterns. Figure 14 shows the reference for the Accum primitive, and Figure 17 exemplifies one of the usage patterns for shape manipulation.

The guardian agent decides whether the output of the proposer violates the affine type constraint and corrects the implementation when necessary. The guardian agent consists of a fused gate and corrector. Figure 20 shows the base

prompt for the guardian agent, which provides input and output examples of variable reuse where the variable is reused zero, one, or two times.

## 4.2. Code generator

After the LLM agents, a code generator takes in the generated implementation and outputs a self-contained pytestable Python script. With this code generator, the LLM agents only need to output implementations without other helper code. We embed the STeP specifications written in natural language to Python (see Figure 21 and Figure 22). Beneath this Python frontend is a functional simulator that calculates the result of STeP programs. Since the essence of each ASPL is the programming abstraction (semantics) instead of its syntax (Liskov, 2011), we choose to prompt LLMs with their familiar Python syntax. In this way, LLMs can focus more on reasoning about the STeP programming abstraction without being distracted by less familiar syntax.

## 4.3. Verifier

Fast verification is vital because it bottlenecks adaptive self-improvement learning. ASPLs further increase this complexity by requiring a simulator for the library–chip co-design process. Simulating STeP as a general dataflow system in Python would be slow because of its high dynamism (Zhang et al., 2024b). Since we only focus on functional correctness in this work, we meticulously limit the level of dynamism to the degree that ML operators require. Consequently, our simulator emulates stream execution using tensor computation with the necessary control flow.

Our system organization contains two verifiers, and the reward function for our system in Algorithm 1 is $r(x, y) = 1$ for task $y$ if answer $x$ passes these two verifiers and $r(x, y) = 0$ otherwise. One verifier checks for functional correctness. Users program PyTorch to express their ML operators, which elicits LLMs' base knowledge of ML algorithms. The verifier compares the execution results of our simulator with the result tensors of the corresponding PyTorch program on a single set of shapes with random input values. The fidelity of our unit test method builds on practice (Jia et al., 2019) and theory (Gulwani & Necula, 2003). The other verifier checks the affine type constraint by performing static analysis on the abstraction syntax tree of the STeP program with the Python ast module (Ronacher, 2008).

## 4.4. Structual intermediate representation

Good interfaces can improve the performance of agentic coding systems (Yang et al., 2024; Wei et al., 2024). Therefore, we are inspired by the intermediate representation (IR) technique from compiler literature (Lattner et al., 2021; Vasilache et al., 2022) and use a structural IR to unify the interfaces of our agentic system. Specifically, this structural IR is the interface between users and the agentic system and between LLM agents and the code generator within the system.

Our structural IR encodes necessary information using a data serialization language. It externalizes and condenses programs instead of simply formatting the prompt without changing the content. Comparing the structural IR in Figure 18 with the equivalent bare Python in Figure 19 for the same task, users only need to state two things: the ML operator to implement and the specialized functions as in Figure 15 without any redundant glue string. Moreover, structural IR saves tokens by reducing redundant prompts, allowing for more demonstrations.

## 5. Benchmark

We construct a set of tasks to measure the adaptive self-improvement agentic system proposed in Section 3 and Section 4. This benchmark should cover a diverse set of popular ML operators and specialized functions. In total, we collect 26 tasks covering 8 groups of ML operators in common LLM model architectures, as shown in Figure 3.

### 5.1. Metric

We choose pass@k (Chen et al., 2021) as the metric for task completion. Pass@k is calculated as Equation (1) given $T$ tasks, $n$ samples of the agentic system, and $c_i$ correct responses for each task $i$. It is useful for us to analyze the metric at two extremes: pass@1 and pass@n. Pass@1 is the expectation of the success rate across tasks. Pass@n is the expectation of the portion of tasks that can be solved given all samples.

$$\text{pass@k} := \frac{1}{T} \sum_{i=1}^{T} \left[ 1 - \frac{\binom{n-c_i}{k}}{\binom{n}{k}} \right] \qquad (1)$$

### 5.2. Benchmark Construction

We construct the benchmark from first principles and do not favor any kind of task. Firstly, the number of tasks in each group is nearly the same as shown in Figure 3. Secondly, the benchmark has an even distribution of difficulties. Table 2 shows that both tasks requiring shape and arithmetic primitives and tasks with and without reused variables distribute fairly evenly.

We provide a reference implementation for each task. These oracle implementations ensure that each task has at least one correct answer. Each task of one type has either different specialized functions for the same operator or different operators with different specialized functions. More details on the benchmark are in Appendix A.2.

# 6. Experiments

Detailed experimental settings are in Appendix A.3. We also benchmark the tasks with OpenAI-o1 in Table 4 but do not include it in the following experiments to control for test-time compute. As described in Section 4.4, all prompts are formatted in YAML because structural prompts generally benefit (He et al., 2024).

## 6.1. Analysis of adaptive self-improvement learning

This section uses the agentic system organization described in Section 4.1 for the best possible base learning capability. Our evaluation provides the following insights:

**The hard-only examples improve performance more effectively than examples mixed with easier ones.** As shown in Figure 7, the Pareto optimal is composed of hard examples (denoted by "H") for all three models. Moreover, hard examples bring the most significant improvement along the learning curve. In some cases, fewer hard examples may perform better than more examples mixed with easier examples. For example, Iter 3 has better performance than Iter 2 for gpt-4o.

**Mixed examples are required to generate better hard-only examples.** In DeepSeek-V3, although $HM_1$ performs the same as $H_1$ and $HME_1$ performs worse than $H_2$ while taking more tokens, $H_2$ would not be discovered without $HM_1$ and $HME_1$. Although the new hard-only examples do not necessarily improve the performance, they can save input tokens, as exemplified by $HM_4$&$H_5$ of gpt-4o and $HM_5$&$H_6$ of llama.

**Adaptive granularity affects token efficiency and peak performance.** If the adaptive granularity $m$ is too small, then easy examples might dilute the difficulty of training data. If $m$ is too large, then exploration steps might be too conservative and thus waste input tokens. Therefore, there is a sweet spot that balances the training data difficulty and cost of input tokens. As shown in Figure 8, $m = 3$ is that spot. $m = 3$ saves $1.07\times$ tokens over $m = 4$ while maintaining performance. Additionally, $m = 3$ improves the performance by $1.5\times$ at a similar input token cost when compared to $m = 1$.

## 6.2. Ablation study of agentic system organization

The experiments below study the base learning capability of agentic systems without the self-improvement process. **The agentic system can discover non-trivial STeP programs.** Surprisingly, the agentic system composes attention operators with over 50% Pass@1 as shown in Figure 13. That means the agentic system can discover online softmax (Milakov & Gimelshein, 2018) and memory-free streaming attention (Sohn et al., 2024b) with specialized functions

| Method | Success diversity | Failure diversity | Overall diversity | Pass@n |
|---|---|---|---|---|
| Single-w/o-IR | 0.32 | 0.47 | 0.41 | 0.46 |
| Single | 0.27 | 0.64 | 0.50 | 0.62 |
| Agent | **0.34** | **0.68** | **0.52** | **0.85** |

*Table 1.* Analysis of the correlation between semantic diversity of answers and the performance. "-w/o-IR" means "without structural IR". Higher values indicate higher semantic diversity.

| Mode | Metric | Method | Once | Reuse | Avg |
|---|---|---|---|---|---|
| Arith | Pass@1 | Single | **0.685** | 0.232 | 0.444 |
| | | Agent | 0.663 | **0.455** | **0.552** |
| | Pass@n | Single | **7**/7 | 5/8 | 12/15 |
| | | Agent | **7**/7 | **8**/8 | **15**/15 |
| Shape | Pass@1 | Single | 0.145 | 0.004 | 0.094 |
| | | Agent | **0.167** | **0.051** | **0.125** |
| | Pass@n | Single | 3/7 | 1/4 | 4/11 |
| | | Agent | **4**/7 | **3**/4 | **7**/11 |
| Avg | Pass@1 | Single | **0.415** | 0.156 | 0.296 |
| | | Agent | **0.415** | **0.320** | **0.371** |
| | Pass@n | Single | 10/14 | 6/12 | 16/26 |
| | | Agent | **11**/14 | **11**/12 | **22**/26 |

*Table 2.* Analysis of improvement brought by the agentic method. "Arith" and "Shape" mean the oracle implementation only involves arithmetic primitives and involves shape manipulation primitives as introduced in Section 2.3, respectively. "Once" and "Reuse" mean all the streams are used once and more than once, respectively.

provided in Figure 16, which is considered challenging for ordinary programmers.

**Our structual IR improves performance by increasing the sample diversity.** Figure 9 shows the efficacy of our structural IR and agentic method. We calculate the semantic diversity of sampled answers. A group of answers is considered to have the same semantics if their abstract syntax trees are isomorphic. We calculate success diversity as the number of semantically different correct answers divided by the total number of correct answers, averaged across all tasks. Failure diversity is calculated in a similar way but for wrong answers. Overall, diversity combines both metrics. As shown in Table 1, Pass@n has a positive correlation with failure, overall diversity, and the complexity of the methods. However, structural IR can hurt success diversity.

**The guardian agent can correct affine type errors but might corrupt the correct answers output by the proposer agent.** As shown in Table 2, the guardian agent effectively corrects the proposer agent, solving 5 extra reuse tasks (Pass@n of Avg-Reuse from 6/12 to 11/12). Notably,

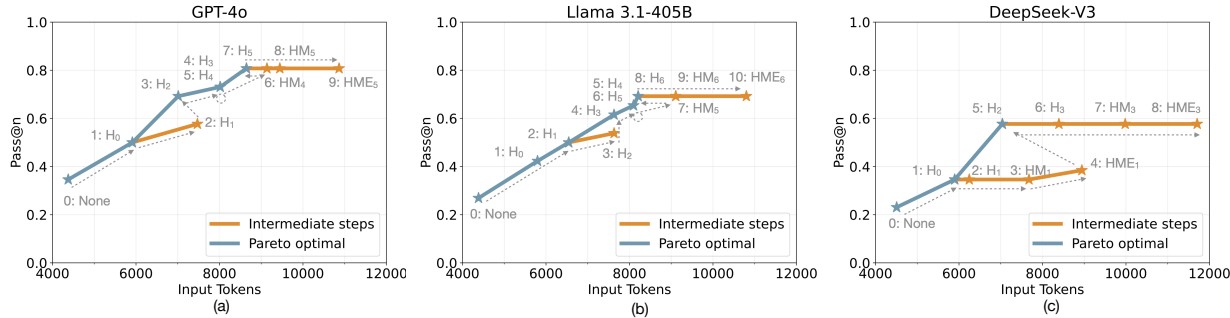

*Figure 7.* Adaptive self-improvement learning improves the agentic system with data generated by itself. The input tokens are averaged across tasks. A task takes increasing input tokens until success or data is used up. "9: $HME_5$" means 9-th iteration, 5-th cycle of adaptive sampling, and demonstrations contain hard (H), medium (M), and easy (E)-earned experiences. "None" means no examples for the first iteration. For Iter $i$, if $Pass@n_{i-1} > Pass@n_{i-2}$, then a new cycle of adaptive sampling starts from "H". Otherwise, the current cycle continues in the order of H→HM→HME. The dash lines are connected in the iteration order. Claude 3.5 Sonnet result is in Figure 11.

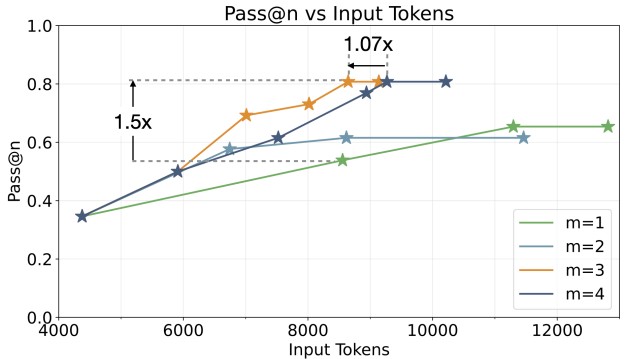

*Figure 8.* Hyperparameter tuning of adaptive granularity $m$ on GPT-4o. This supports using m=3 for Figure 7.

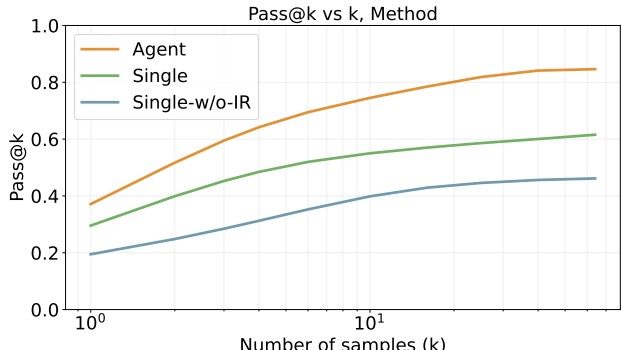

*Figure 9.* Pass@k against the number of samples for a single LLM without structural IR, baseline single LLM, and agentic system.

all the Arith-Reuse tasks can be solved by the agentic system (Pass@n of Arith-Reuse from 5/8 to 8/8). Surprisingly, the agentic system also helps with non-reuse tasks (Pass@n of Shape-Once from 3/7 to 4/7). However, the agentic system reduces the Pass@1 of Arith-Once from 0.685 to 0.663, implying that the guardian agent might corrupt the proposer's output. The agentic system can compensate for such corruption by finishing more tasks, resulting in an unchanged Pass@1 of Avg-Once.

# 7. Related Work

This work introduces a self-improvement agentic system for ASPL code generation, enhancing LLM effectiveness through an adaptive learning algorithm and deliberate agentic system design. Accordingly, we focus the related work on approaches that leverage LLMs for self-improvement and agentic systems on specialized tasks.

## 7.1. Self-improvement learning for LLMs

Self-improvement learning for LLMs typically involves two stages: scoring generated samples (trajectories) and incorporating those samples to enhance the model. Scoring can be achieved through human labeling (Cobbe et al., 2021; Lightman et al., 2023) or through automated methods such as verifiers and heuristics (Wang et al., 2024; Singh et al., 2023). Our method stands out in this context by utilizing AST analysis, offering a more interpretable approach to scoring. When it comes to incorporating samples, models may rely on retrieval (Zhao et al., 2024; Park et al., 2023), reflection (Shinn et al., 2024; Liu et al., 2023; Yao et al., 2023), or reward feedback (Opsahl-Ong et al., 2024; Fernando et al., 2023). Our method introduces a new mechanism in this stage by adaptively extending and prioritizing high-scoring samples.

Our method shares similar reinforcement principles with self-improvement learning at the post-training stage (Bai

et al., 2022; Gulcehre et al., 2023; Tian et al., 2024) but is rewarded at a task-level granularity instead of token-level. Specifically, each action is a program implementation instead of token prediction and the state is defined by earned experiences rather than generated sequences.

### 7.2. Agentic system organization for specialized tasks

Task-specific organization has proven effective in enhancing the performance of agentic systems across diverse coding tasks (Zhang et al., 2024c; Fang et al., 2024; Guan et al., 2024). We adopt an agentic system organization specifically for ML library development using an ASPL. Such domain knowledge can be further augmented with automatic agentic system design tools (Khattab et al., 2023; Hu et al., 2024; Zhang et al., 2024a). Furthermore, well-designed interfaces between agents, tools, and other agents have been shown to improve performance (Schick et al., 2023; Yang et al., 2024; Wu et al., 2023), and a structural IR enables these interfaces to be highly task-aligned in our system.

## 8. Conclusion

ML library development using ASPLs is a critical component of the ML ecosystem, but it remains poorly automated. To address this limitation, we co-design the learning process and agentic system around a central objective: enabling complex reasoning with limited data. Our methods simultaneously implement non-trivial ML operators and produce a self-improving agent. Consequently, our system not only supports experts in developing ML libraries but also offers valuable resources for other systems. We recognize that the challenges of complex reasoning under limited data extend beyond this domain and envision applying similar self-improvement techniques with LLM agents to a broad class of tasks.

## Impact Statement

Our work on adaptive self-improvement learning LLM agentic systems for automating ML library development using ASPLs has several potential societal implications. The primary impact is the potential to significantly enhance the productivity of ML library developers, which could accelerate the development of more efficient ML systems. This advancement could democratize access to ML development tools and reduce the technical barriers to entry in the field. Additionally, our technique offers an approach for deploying LLM agents in scenarios requiring complex reasoning with limited data availability. While these developments primarily aim to advance the field of Machine Learning, we acknowledge that increased automation in software development could impact the nature of programming work and skills required in the field. We believe these poten-

tial implications warrant ongoing discussion and careful consideration as the technology develops.

## Acknowledgement

We thank Vishnu Sarukkai, Haochen Shi, Xinhao Li, Gina Sohn, Nathan Zhang, Tian Zhao, Yu Sun, Anjiang Wei, Chun Deng, Qizheng Zhang, Rubens Lacouture, Fredrik Kjolstad and the anonymous reviewers for their feedback on this paper. This research was supported in part by the Pervasive Parallelism Lab Affiliates Program Fund. Any opinions, findings, and conclusions or recommendations expressed in this material are those of the authors and do not necessarily reflect the views of the aforementioned funding agencies.

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

# A. Appendix

## A.1. STeP introduction

$$\sum_{i=0}^{1} G_i \cdot \text{gelu}(W_i X) \text{ with } N_i = \mathbf{I}[G_i > 0] \tag{2}$$

---

**Algorithm 2** STeP for a simplified MoE module

---

**input**  $X$: [m,n] of Buffer(k), $N$: [m,n] of Multihot(e), $G$:
    [m,n] of Buffer(e)
**output**  [m,n] of Buffer(k)
**param**  $W_0$: [k,d], $W_1$: [k,d]
**func** weightedsum                  ▷ external function
    **type:** Buffer(k) $\rightarrow$ $\langle$Buffer(k), Scalar$\rangle$ $\rightarrow$ Buffer(k)
    **fn** (s,v) = s + $v_0 * v_1$
    **init:**  s = 0
**func** expert$_0$
    **type:** $\langle$Buffer(k), Buffer(e)$\rangle$ $\rightarrow$ $\langle$Buffer(k), Scalar$\rangle$
    **fn** (v) = gelu($W_0 v_0$), $v_1[0]$)
**func** expert$_1$
    **type:** $\langle$Buffer(k), Buffer(e)$\rangle$ $\rightarrow$ $\langle$Buffer(k), Scalar$\rangle$
    **fn** (v) = gelu($W_1 v_0$), $v_1[1]$)
  1: $S_0 = \text{Zip}(X, G)$
  2: $S_1^0, S_1^1 = \text{Copy}(N)$
  3: $S_2 = \text{Partition}(2, S_0, S_1^0)$
  4: $S_3 = [\text{Map}(\text{expert}_0, S_2[0]), \text{Map}(\text{expert}_1, S_2[1])]$
  5: $S_4 = \text{Merge}(\text{weightedsum}, S_3, S_1^1)$

---

*Figure 10.* Algorithm 2 is a STeP program example for Equation (2). The type signature follows the Haskell style where $\rightarrow$ connects a sequence of argument types with one return type. Three **func**s are external functions provided by the hardware.

As shown in Figure 10, Copy duplicates a stream for the affine type constraint. Zip combines two streams of values into a stream of tuples. Map applies the function (**fn**) on each input value. Partition routes tokens of the data stream to experts assigned by the index stream ($S_1^0$). Merge accumulates tokens from experts. The accumulation of Merge is parameterized by **fn** and **init** where **init** initializes the state and **fn** updates the state with the input value. Partition and Merge are used in pairs, sharing the same index stream. $\langle\rangle$ represents the Tuple type of value tokens. Buffer, Multihot, and Scalar are also types of value tokens, parametrized by the generic data type like float and half, which is omitted for simplicity in the algorithm. Buffer and Multihot types are further parametrized by the shape in the paratheses. Shape manipulation primitives include `Promote`, `Repeat` and `RepeatRef`. Figure 4(a) also shows two streams. `S1` is a control token signaling the end of rank-1. `01` is a value token of multihot vector type served in index streams. (`v0`,`v1`) is a value token of type Tuple(Scalar, Reference) because the weight and input streams are composed of scalar and reference values, respectively.

## A.2. Benchmark details

As shown in Figure 3, Group 4 tasks have the same operator: $softmax(S) \cdot V$ where $S$ equals $QK^T$. They differ in external functions as shown in Figure 16. Group 4 can use RDA's on-chip fusion to compose scale-dot-product attention with Group 5. Group 7 also differs in external functions. Group 5 contains three dataflow orders: inner-product("$mnk, mdk \rightarrow mnd$"), row-wise("$mnk, mkd \rightarrow mnd$"), and outer-product("$mkn, mkd \rightarrow mnd$"). Group 6 contains GptJ and NeoX styles which differ in pairing even and odd or the first and second half positions (vLLM, 2023). Group 8 contains LayerNorm and RMSNorm.

The last three in Table 3: index, expert, and etoe all come from MoE. MoE contains token (Shazeer et al., 2017) Equation (3) and expert choice routing (Zhou et al., 2022) Equation (4).

$$S = softmax(X \cdot W_g), S \in \mathbb{R}^{n \times e}$$
$$G, I = TopK(S) \qquad\qquad \text{Along expert dimension} \tag{3}$$

$$S = softmax(X \cdot W_g), S \in \mathbb{R}^{n \times e}$$
$$G, I = TopK(S^T) \qquad\qquad \text{Along token dimension} \tag{4}$$

The expert choice routing can have another MLP auxiliary predictor for causal inference when it is a binary choice (Raposo et al., 2024) Equation (5).

$$S = \sigma((gelu(X \cdot W_{g_0})) \cdot W_{g_1}), S \in \mathbb{R}^{n \times 1} \quad \text{expert=2}$$
$$G, I = S > 0.5 \tag{5}$$

| Group | Description | Count |
|--------|-------------|-------|
| attn | Softmax(S)@V part | 3 |
| gemm | Matrix multiplication with expanded reduction dimension | 3 |
| bmm | Batch matrix-matrix product | 3 |
| norm | RMSNorm and LayerNorm (without bias and gain) | 3 |
| rope | GptJ and NeoX style of RoPE | 3 |
| index | Index generation of MoE router | 4 |
| expert | Expert execution of MoE | 3 |
| etoe | End-to-end of MoE module | 4 |

*Table 3.* Description of each type of tasks. Matmul is short for matrix multiplication.

### A.3. Experiment settings

In Section 6.1 we sample 64 times for each temperature of 0.4, 0.7, and 1.0, recording the best result. In Section 6.2, we sample 64 times at temperature 0.7 on Claude 3.5 Sonnet to control variables. Four models are: claude-3-5-sonnet-20241022 of Anthropic API (Claude 3.5 Sonnet), gpt-4o-2024-11-20 of OpenAI API (GPT-4o), deepseek-chat of DeepSeek API (DeepSeek-V3), and Meta-Llama-3-1-405B-Instruct-Turbo of TogetherAI API (Llama 3.1-405B). Maximum output tokens are set as 1024 and the seed for GPT-4o is 42.

### A.4. Additional experiments

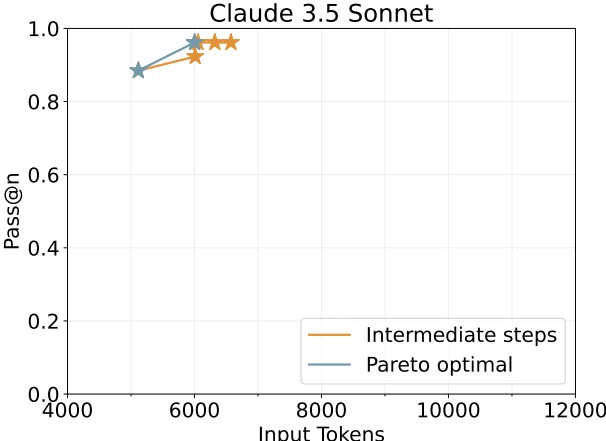

*Figure 11.* Self-improvement learning curve with m=3. Claude 3.5 Sonnet consumes much fewer tokens than other models because the input tokens are counted by the least necessary number of tokens averaged across tasks. A task can take increasing input tokens but still fail. Sonnet only has 3 unsolved tasks left. Therefore, although it has the most example tokens, the average number of input tokens across tasks is still less than others.

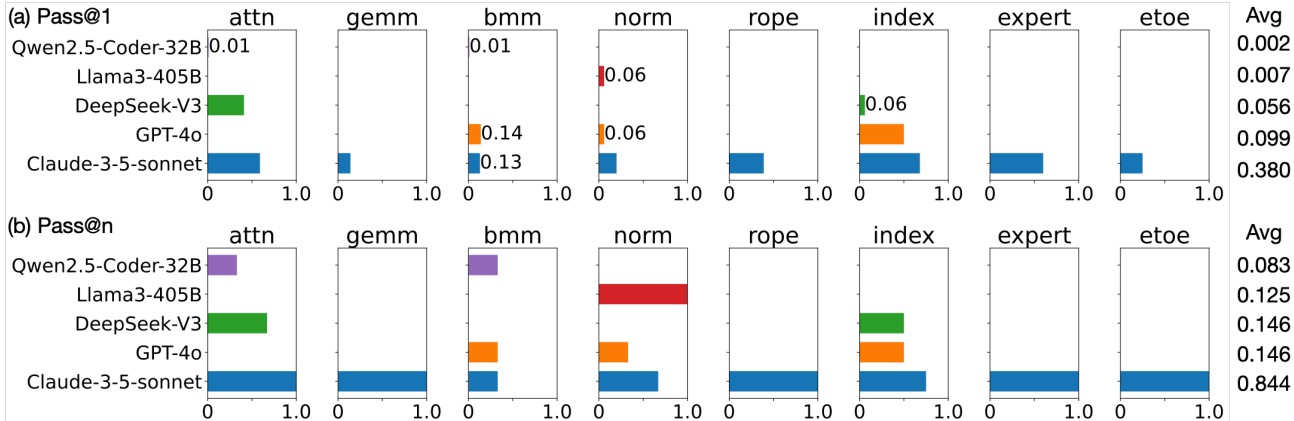

*Figure 12.* Result of five models at temperature 0.7.

As shown in Table 4, OpenAI-o1 achieves similar performance at the cost of more tokens than a single Claude-3-5-Sonnet. This observation aligns with previous findings that scaling pretraining is preferable over inference for challenging tasks (Snell et al., 2024). Meanwhile, a single Claude-3-5-Sonnet proposer can finish more tasks than the OpenAI-o1 using fewer tokens. Table 3 contains abbreviations for all tasks.

Our system completed each task in under 10 minutes on average. In contrast, during our pilot study, a domain expert was

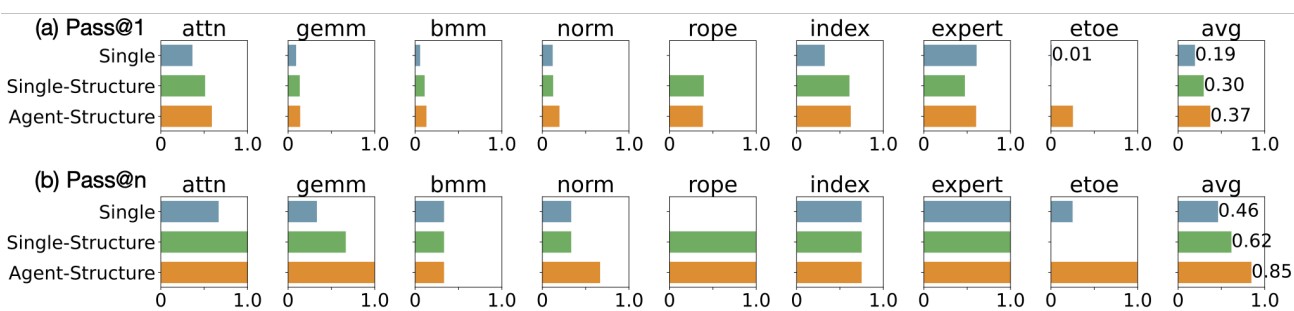

*Figure 13.* Result of three methods on Claude 3.5 Sonnet at temperature 0.7.

| Model | Metric (↑) | attn | gemm | bmm | norm | rope | index | expert | etoe | avg |
|---|---|---|---|---|---|---|---|---|---|---|
| Qwen2.5-Coder-32B | Pass@1 | 0.010 | 0 | 0.052 | 0 | 0 | 0.004 | 0 | 0 | 0.008 |
| | Pass@n | 0.33 | 0 | **0.33** | 0 | 0 | 0.25 | 0 | 0 | 0.115 |
| Llama3-405B | Pass@1 | 0.010 | 0 | 0 | 0.057 | 0.089 | 0.016 | 0 | 0 | 0.020 |
| | Pass@n | 0.33 | 0 | 0 | **1.00** | 0.67 | 0.25 | 0 | 0 | 0.269 |
| DeepSeek-V3 | Pass@1 | 0.438 | 0 | 0 | 0 | 0 | 0.113 | 0 | 0 | 0.068 |
| | Pass@n | **1.00** | 0 | 0 | 0 | 0 | **0.75** | 0 | 0 | 0.231 |
| GPT-4o | Pass@1 | 0.021 | 0.016 | **0.214** | 0.094 | 0 | 0.258 | 0.005 | 0 | 0.080 |
| | Pass@n | 0.33 | 0.67 | **0.33** | 0.67 | 0 | 0.5 | 0.33 | 0 | 0.346 |
| Claude-3-5-sonnet | Pass@1 | **0.620** | **0.229** | 0.146 | **0.208** | **0.526** | **0.676** | **0.688** | **0.324** | **0.433** |
| | Pass@n | **1.00** | **1.00** | 0.33 | **1.00** | **1.00** | **0.75** | **1.00** | **1.00** | **0.885** |
| OpenAI-o1 (n=8) | Pass@1 | 0.208 | 0.042 | 0 | 0 | 0.083 | 0.343 | 0.583 | 0 | 0.159 |
| | Pass@n | 0.67 | 0.33 | 0 | 0 | 0.67 | 0.5 | **1.00** | 0 | 0.385 |

*Table 4.* The performance of the self-improvement agentic system across models.

unable to write a single program within 48 hours, as they had to do trial-and-error and accumulate experience sequentially. Our system, by comparison, can perform these explorations in parallel. In the future, collaboration with HCI researchers will enable more extensive experiments comparing time and effort required by human programmers versus the system, providing quantitative data on usability and cognitive load.

Differences in tool access and computational load might have influenced the outcomes. Therefore, we conducted an experiment that aligned both aspects.

For computation fairness, we matched the token count of the single model with the agent and self-improved models by resampling. All model variants (single, agent, self-improved) have access to the same verifier so it is fair; the difference lies in how the verifier is leveraged. Self-improved models incorporate it throughout the process, while others use it only at the end as a final judge. We chose Claude-3-5-Sonnet and GPT-4o as base models. As shown in Table 5, our agentic systems still perform better under this fair setting.

| Pass@n | Claude Sonnet Single | Claude Sonnet Agent & Self-improved | GPT-4o Single | GPT-4o Agent & Self-improved |
|---|---|---|---|---|
| From | 0.73 | 0.73 | 0.23 | 0.23 |
| To | 0.77 | **0.96** | 0.38 | **0.81** |

*Table 5.* Performance comparison of single models vs agent & self-improved models with token count matching.

We conducted supervised finetuning (SFT) using GPT-4o. Table 6 shows that SFT can improve performance, but less than our self-improvement approach.

Since we do not know the exact SFT algorithm of OpenAI service for FLOPs matching, we tried our best to favor the SFT method. We began with the same 133 correct samples from all completed tasks used in the first iteration of self-improvement. Different from self-improvement which only picks 1 correct program per completed task, we picked all 133 programs to form the training dataset of SFT. We created three SFT datasets with varying prompt compositions:

- 133 (base prompt+question+answer)

- 133 (question+answer)

- 17 (question+answers deduplicated via AST)

Each dataset was used to train a separate SFT model. After that, we sampled each model on all the uncompleted tasks.

| Pass@n | Finetuned | Self-improved |
|--------|-----------|---------------|
| From   | 0.35      | 0.35          |
| To     | 0.62      | **0.81**      |

*Table 6.* Performance comparison between finetuned and self-improved Models

We assessed code maintainability using two metrics: maintenance index without comments (MIwoc) and with comments (MI) [2]. The comment weight (MIcw) is defined as MI - MIwoc and falls in [0, 49]; MI > 85 indicates good maintainability. Using all correct programs from our best model (self-improved agentic Claude Sonnet), we recorded the top MIwoc and MI per task. The mean MIwoc is 102, MI is 149, and MIcw is 47—indicating well-commented, maintainable code.

As shown in Table 7, we used the same set of programs as the maintainability statistics to measure complexity. We measured these three metrics:

- Lines of code: Counted via primitive calls (excluding comments/blank lines)

- Shape transformations: Counted by use of Promote, Repeat, RepeatRef, and Flatten primitives

- Specialized instructions: Counted as the number of specialized functions in task descriptions

| Metric | Min | Max | Mean |
|--------|-----|-----|------|
| Lines of code | 4 | 17 | 8.67 |
| Shape transformations | 0 | 6 | 1.13 |
| Specialized instructions | 2 | 7 | 3.68 |

*Table 7.* The assessment of complexity across all completed tasks.

### A.5. Generalization

The proposed system can be extended to other scenarios that require complex reasoning with limited example data and well-defined evaluation metrics. We outline the general recipe below.

As shown in Figure 1, the agentic system organization is constructed in three main steps. First, system designers define the format of both the task and its expected output. Once the format is specified, the next step is to build a verifier for the task. With the format and verifier in place, designers can either use a single LLM or design LLM agents tailored to the

---

[2]https://www.verifysoft.com/en_maintainability.html

domain—similar to how we handle the type constraints of the STeP language. After completing these three steps, the task can be handed over to our system, which will automatically carry out adaptive self-improvement learning.

The adaptive self-improvement learning system also exposes several tunable hyperparameters which will be helpful when the results are not satisfactory. The most direct control is the number of parallel sampling. Users can also adjust the adaptive granularity parameter m for experience stratification. Additionally, domain-specific filtering heuristics—such as the minimal code length heuristic used by us—can be incorporated to further guide the learning process.

Our approach has two parts: adaptive self-improvement learning and agentic system organization. The learning process is broadly applicable to other programming languages; the challenges lie in tailoring agentic systems to other languages.

Mainstream languages like CUDA, HIP, and CPU vector intrinsics exhibit global properties such as arbitrary memory access, data layout sensitivity, and side effects. Similarly, STeP enforces a global affine type constraint. Our framework addressed this using a *guardian agent* that detects and corrects affine type violations. This concept generalizes: domain-specific guardian agents can monitor and enforce global properties of various languages, adapting the STeP solution more broadly.

A second challenge is that LLMs may lean toward surface-level patterns in mainstream languages due to their existence in training data, potentially missing more optimal or novel transformations. As shown in Section 6.2, our structural IR can increase sample diversity and thus boost the LLM agentic system performance. Extending this, structural IRs and tailored code generators can guide LLMs toward more creative solutions beyond conventional patterns.

We also conducted an experiment on the AIME-2024 dataset [3] which contains 30 challenging problems from the American Invitational Mathematics Examination (AIME) 2024. We applied our adaptive self-improvement learning to the Claude-3-5-Sonnet base model and increased Pass@n from 0.50 to 0.67. This demonstrates the potential capabilities of our system on other tasks.

### A.6. Additional explanation

| Abbv. | Description |
|---|---|
| DSA | Domain Specific Architectures |
| DSL | Domain Specific Language |
| ASPL | Architecture Specific Programming Language |
| RDA | Reconfigurable Dataflow Architecture, a DSA for AI |
| STeP | Streaming Tensor Program, an ASPL for next-generation RDA |

*Table 8.* Explanation of the abbreviations.

Existing auto-tuning code generation systems like Spiral (Puschel et al., 2005), Ansor (Zheng et al., 2020), and AKG (Zhao et al., 2021) rely on predefined optimization strategies and cost models. On the contrary, we employ LLMs to generate and refine code for machine learning libraries, particularly targeting ASPLs. We explore the shift towards utilizing the reasoning capabilities of LLMs for code generation and optimization, enabling the system to adapt and improve over time without explicit human intervention.

---

[3]https://huggingface.co/datasets/Maxwell-Jia/AIME_2024

## A.7. Code and prompt examples

```
- name: Accum
  desc: |
  Accum is a primitive operation that applies a function to a stream in a recursive manner.
  The function is applied to the first element of the stream and the initial state to produce the first output
      element.
  The function is then applied to the second element of the stream and the output of the previous application to
      produce the second output element, and so on.
  The state is initialized at rank `b` of the input stream. The output stream's shape is the input stream's shape
      excluding the first `b` dimensions.

  examples:
  - inputs:
    - name: E0
      dtype: fp32
      dims: [M, N]
      data_gen: torch.rand
    fns:
    - name: Sum
      apply: |
        return [state[0] + input[0]]
      init: [0]
      input_dtype: fp32
      output_dtype: fp32
      func_name: fn_sum
    outputs:
    - name: S0
      dtype: fp32
      dims: [N]
      data_transform:
        - |
        torch.sum(input_data['E0'], 1, keepdim=False)
    impl: |
      E1 = step.Accum(fn=fn_sum, b=1).apply(E0)
      return E1
```

*Figure 14.* The reference of `Accum` that contains the definition and an example. Each example in the examples field is composed of task description and implementation.

```
inputs:
  - name: E0
    dtype: fp32
    dims: [M, N]
    data_gen: torch.rand
  - name: E1
    dtype: Buffer(fp32, [D])
    dims: [M, N]
    data_gen: torch.rand

fns:
  - name: MaxSum
    apply: |
      m_t, l_t, o_t = state # scalar, scalar, [D]
      s_t, v_t = input # scalar, [D]
      m_next = torch.max(m_t, s_t) # scalar
      l_prim_t = torch.exp(m_t - m_next) * l_t
      p_t = torch.exp(s_t - m_next)
      l_next = p_t + l_prim_t
      o_next = l_prim_t * o_t / l_next + p_t * v_t / l_next
      return [m_next, l_next, o_next]
    init: [-inf, 0, 0]
    input_dtype: [fp32, "Buffer(fp32, [D])"]
    output_dtype: [fp32, fp32, "Buffer(fp32, [D])"]
    func_name: fn_maxsum

  - name: GetThird
    apply: |
      return [input[2]]
    input_dtype: [fp32, fp32, "Buffer(fp32, [D])"]
    output_dtype: Buffer(fp32, [D])
    func_name: fn_getthird

outputs:
  - name: S0
    dtype: fp32
    dims: [D, N]
    data_transform:
      - |
        torch.bmm(torch.softmax(input_data['E0'], 1).unsqueeze(1), input_data['E1']).squeeze(1)

impl: |
```

*Figure 15.* Task description of attn task 0 in our structural IR, where the LLM needs to complete `impl`.

```
# Task 0
# MaxSum
def apply(state, input):
    m_t, l_t, o_t = state
    s_t, v_t = input
    m_{t+1} = max(m_t, s_t)
    l'_t = l_t * e^{(m_t - m_{t+1})}
    p_t = e^{(s_t - m_{t+1})}
    l_{t+1} = p_t + l'_t
    o_{t+1} = 1/l_{t+1} (l'_t * o_t + p_t * v_t)
    return (m_{t+1}, l_{t+1}, o_{t+1})

def init():
    return (-∞, 0, 0)

# GetThrid
def apply(input):
    return input[2]
```

```
# Task 1
# ExpMaxDiff
def apply(state, input):
    m_t, e_t, d_t = state
    s_t, = input
    m_{t+1} = max(m_t, s_t)
    Δm = m_t - m_{t+1}
    e_{t+1} = e^{(s_t - m_{t+1})}
    d_{t+1} = e^{Δm}
    return (m_{t+1}, e_{t+1}, d_{t+1})

def init():
    return (-∞, 0, 0)

# DivSum
def apply(state, input):
    v_t, e_t, d_t = input
    l_t, o_t = state
    l'_t = d_t * l_t
    l_{t+1} = e_t + l'_t
    o_{t+1} = 1/l_{t+1} (l'_t * o_t + e_t * v_t)
    return (l_{t+1}, o_{t+1})

def init():
    return (0, 0)

# GetSecondThrid
def apply(input):
    return input[1], input[2]

# GetSecond
def apply(input):
    return input[1]
```

```
# Task 2
# ExpMaxDiff
def apply(state, input):
    m_t, e_t, d_t = state
    s_t, = input
    m_{t+1} = max(m_t, s_t)
    Δm = m_t - m_{t+1}
    e_{t+1} = e^{(s_t - m_{t+1})}
    d_{t+1} = e^{Δm}
    return (m_{t+1}, e_{t+1}, d_{t+1})

def init():
    return (-∞, 0, 0)

# GetSecondThrid
def apply(input):
    return input[1], input[2]

# WeightedSumSingle
def apply(state, input):
    e_t, d_t = input
    r_t = state
    return (r_t * d_t + e_t)

def init():
    return 0

# WeightedSumDouble
def apply(state, input):
    v_t, e_t, d_t = input
    return (state * d_t + e_t * v_t)

def init():
    return 0

# Div
def apply(input):
    r_t, l_t = input
    return l_t / r_t
```

*Figure 16.* Inner functions for 3 tasks of attn. Task 0 encapsulates the whole innermost loop body of FlashAttention (Dao et al., 2022) in the `MaxSum` function. Task 1 splits the `MaxSum` into `ExpMaxDiff` and `DivSum`. Task 2 postpones the division of summation as in FlashAttention2 (Dao, 2023). The bold symbols are streams with type 1D Buffer, and the symbols are streams with type Scalar.

```
name: Stashing dimension
desc: |
  When the pritmives require a non-one dimension to be inserted as a non-innermost dimension, a Bufferize&Streamify
      pair can wrap the primitives to adjust the dimension.
  This pattern is useful for Repeat and RepeatRef primitives.
examples:
  - inputs:
    - name: E0
      dtype: fp32
      dims: [M, N, K]
      data_gen: torch.rand
    outputs:
    - name: S0
      dtype: fp32
      dims: [M, N, D, K]
      data_transform:
        - |
          input_data['E0'].unsqueeze(1).repeat(1, D_value, 1, 1)
    impl: |
      E1 = step.Bufferize(a=2).apply(E0) # E1: {dtype: Buffer(fp32, [M, N]), dims: [K]}
      E2 = step.Repeat(n=D).apply(E1) # E2: {dtype: Buffer(fp32, [M, N]), dims: [D, K]}
      E3 = step.Streamify().apply(E2) # E3: {dtype: fp32, dims: [M, N, D, K]}
      return E3
```

*Figure 17.* An example of usage pattern that contains 3 shape manipulation primitives: Bufferize, Repeat, and Streamify.

```yaml
inputs:
  - name: E0
    dtype: fp32
    dims: [M, N]
    data_gen: torch.rand
  - name: E1
    dtype: Buffer(fp32, [D])
    dims: [M, N]
    data_gen: torch.rand

fns:
  - name: MaxSum
    apply: |
      m_t, l_t, o_t = state # scalar, scalar, [D]
      s_t, v_t = input # scalar, [D]
      m_next = torch.max(m_t, s_t) # scalar
      l_prim_t = torch.exp(m_t - m_next) * l_t
      p_t = torch.exp(s_t - m_next)
      l_next = p_t + l_prim_t
      o_next = l_prim_t * o_t / l_next + p_t * v_t / l_next
      return [m_next, l_next, o_next]
    init: [-inf, 0, 0]
    input_dtype: [fp32, "Buffer(fp32, [D])"]
    output_dtype: [fp32, fp32, "Buffer(fp32, [D])"]
    func_name: fn_maxsum

  - name: GetThird
    apply: |
      return [input[2]]
    input_dtype: [fp32, fp32, "Buffer(fp32, [D])"]
    output_dtype: Buffer(fp32, [D])
    func_name: fn_getthird

outputs:
  - name: S0
    dtype: fp32
    dims: [D, N]
    data_transform:
      - |
        torch.bmm(torch.softmax(input_data['E0'], 1).unsqueeze(1), input_data['E1']).squeeze(1)

impl: |
  E3 = step.Zip().apply((E0, E1))
  E4 = step.Accum(fn=fn_maxsum, b=1).apply(E3)
  E5 = step.Map(fn=fn_getthird).apply(E4)
  E2 = step.Streamify().apply(E5)
  return E2
```

*Figure 18.* The complete implementation of attn task 0 written in structural IR.

```python
import step
from sympy import Symbol
import torch

M = Symbol("M")
N = Symbol("N")
K = Symbol("K")
D = Symbol("D")
M_value = 5
N_value = 7
K_value = 9
D_value = 11
ctx = {M: M_value, N: N_value, K: K_value, D: D_value}
input_dtype = {'E0': step.Scalar("float"), 'E1': step.Buffer(step.Scalar("float"), [D])}
input_data = {'E0': torch.rand(N_value, M_value), 'E1': torch.rand(N_value, M_value, D_value)}

class MaxSum(step.Fn):
    def __init__(self, input, output):
        super().__init__("MaxSum", input, output)
    def getInit(self):
        return [torch.tensor(float('-inf')), torch.tensor(0), torch.zeros(D_value)]
    def apply(self, state, input):
        m_t, l_t, o_t = state # scalar, scalar, [D]
        s_t, v_t = input # scalar, [D]
        m_next = torch.max(m_t, s_t) # scalar
        l_prim_t = torch.exp(m_t - m_next) * l_t
        p_t = torch.exp(s_t - m_next)
        l_next = p_t + l_prim_t
        o_next = l_prim_t * o_t / l_next + p_t * v_t / l_next
        return [m_next, l_next, o_next]
fn_maxsum = MaxSum(step.STuple((step.Scalar("float"), step.Buffer(step.Scalar("float"), [D]))),
    step.STuple((step.Scalar("float"), step.Scalar("float"), step.Buffer(step.Scalar("float"), [D]))))

class GetThird(step.Fn):
    def __init__(self, input, output):
        super().__init__("GetThird", input, output)
    def apply(self, input):
        return [input[2]]
fn_getthird = GetThird(step.STuple((step.Scalar("float"), step.Scalar("float"), step.Buffer(step.Scalar("float"),
    [D]))), step.Buffer(step.Scalar("float"), [D]))

def prepare():
    E0 = step.Stream("E0", step.Scalar("float"), 1, [M, N])
    E0.ctx = ctx
    E0.data = [input_data['E0']]
    E1 = step.Stream("E1", step.Buffer(step.Scalar("float"), [D]), 1, [M, N])
    E1.ctx = ctx
    E1.data = [input_data['E1']]
    return E0, E1

def check_shape(S0):
    assert S0.shape == [D, N]
    assert S0.dtype == step.Scalar("float")

def check_data(S0):
    S0_data_0 = torch.bmm(torch.softmax(input_data['E0'], 1).unsqueeze(1), input_data['E1']).squeeze(1)
    torch.testing.assert_close(S0.data[0], S0_data_0)

def test():
    E0, E1 = prepare()
    S0 = body(E0, E1)
    check_shape(S0)
    check_data(S0)

def body(E0, E1):
    E3 = step.Zip().apply((E0, E1))
    E4 = step.Accum(fn=fn_maxsum, b=1).apply(E3)
    E5 = step.Map(fn=fn_getthird).apply(E4)
    E2 = step.Streamify().apply(E5)
    return E2
```

*Figure 19.* The unit test of the implementation of attn task 0 in Python produced by the code generator from structural IR shown in Figure 18.

```
desc: |
  Streaming Tensor Programs (STeP) provides a higher-level abstraction for dataflow systems.
  The streams can be only consumed once. Your task is to use Copy primitives to create a new stream that is a copy
      of the input stream when necessary.

examples:
  - input_impl: |
      E2 = step.Partition(N=E_value).apply((E0, E1))
      E3 = [step.Map(fn=fn).apply(s) for fn, s in zip(matmul_fns, E2)]
      E4 = step.Merge(fn=fn_sum).apply((E3, E1))
      return E4

    output_impl: |
      E1_0, E1_1 = step.Copy().apply(E1)
      E2 = step.Partition(N=E_value).apply((E0, E1_0))
      E3 = [step.Map(fn=fn).apply(s) for fn, s in zip(matmul_fns, E2)]
      E4 = step.Merge(fn=fn_sum).apply((E3, E1_1))
      return E4

    explanation: |
      Stream E1 is consumed twice in the input implementation. To ensure that the stream is consumed only once, we
          create a copy of the stream E1 and use the copy in the second step.

  - input_impl: |
      E1 = step.Map(fn=fn_predict).apply(E0)
      E2 = step.Map(fn=fn_router).apply(E1)
      E3 = step.Map(fn=fn_affinity).apply(E0)
      E4 = step.Zip().apply((E0, E3))
      return E4

    output_impl: |
      E0_0, E0_1 = step.Copy().apply(E0)
      E0_2, E0_3 = step.Copy().apply(E0_0)
      E1 = step.Map(fn=fn_predict).apply(E0_1)
      E2 = step.Map(fn=fn_router).apply(E1)
      E3 = step.Map(fn=fn_affinity).apply(E0_2)
      E4 = step.Zip().apply((E0_3, E3))
      return E4

    explanation: |
      Stream E0 is consumed 3 times in the input implementation. To ensure that all streams are consumed only once,
          we create a copy of the stream E0 and use the copy in the subsequent steps.

  - input_impl: |
      E1 = step.Bufferize(a=1).apply(E0)
      E2 = step.Map(fn=fn_gate).apply(E1)
      E3 = step.Map(fn=fn_top2).apply(E2)
      return E3, E2

    output_impl: |
      E1 = step.Bufferize(a=1).apply(E0)
      E2 = step.Map(fn=fn_gate).apply(E1)
      E3 = step.Map(fn=fn_top2).apply(E2)
      return E3, E2

    explanation: |
      All streams are consumed only once in the input implementation. No need to create a copy of any stream.
```

*Figure 20.* Base prompt for the guardian agent.

```
### <a name="Accum"></a>Accum
Accumulate the lower `b` dimensions in `Stream<A,a>` into a single value of type `B`. **Accum** will continue to
    dequeue and accumulate to a value of type `B` by calling the given accumulation function (`Fn(A,B)->B`) until
    it sees a `.Sb` in the input stream. Then, it will emit the accumulated value of type `B` into the output
    stream and initialize the accumulator with the given initialize function.
```
```
Accum<A,B,a,b>: Fn(A,B) -> B, Fn() -> B, Stream<A,a> -> Stream<B,a-b>
                (accumulate)  (initialize)
Precondition: 0 < b <= a
```
```
We can think of `b` as the minimum stop token level **Accum** has to see before emitting the accumulated values.
    More details on how to set `b` according to the type of reduction we do can be found in the below examples.

<details>

<summary>
Examples
</summary>

**Example1: Rowmax**  

```
```
Goal: [B,N,E] -> [B,N] (Reduce over the inner-most dim)
Accum<A=f32, B=f32, a=3, b=1>:
  Fn(f32,f32)->f32, Fn()->f32, Stream<f32,3>->Stream<f32,2>

Precondition: 0 < b <= a
                (=1)  (=3)

```
```
We will call the given function (max) on every dequeue and emit the accumulated value when we see a `.Sx(x>=b)`. b
    is 1 in this example because we have to see the whole vector to obtain the reduced value.

```

*Figure 21.* The specification of `Accum` primitive in the STeP document.

```python
class Accum(OpBase):
    def __init__(self, **kwargs):
        super().__init__("Accum", **kwargs)

    def apply(self, input: base.Stream, name=""):
        b = self.config["b"]
        fn: base.Fn = self.config["fn"]
        assert isinstance(fn, base.Fn), f"Accum should take one of provided fns as input, but get {type(fn)}"
        assert fn.input == input.dtype, f"Accum should take {fn.input} as input, but get {input.dtype}"
        assert b > 0 and b <= input.rank, f"Accum should take a positive integer b less than or equal to the rank
             of the input, but get b: {b} and input rank: {input.rank}"

        result = base.Stream(
            self.getName(name), fn.output, input.rank - b, input.shape[b:]
        )
        if input.data is not None:
            result.ctx = input.ctx
            # TODO: Construct a general application function here
            output_indices = get_full_indices(base.subsOuterShape(result.shape, result.ctx))
            input_indices = get_full_indices(base.subsOuterShape(input.shape[:b], input.ctx))
            if isinstance(result.dtype, base.Element) or isinstance(result.dtype, base.Buffer):
                output_shapes = [base.subsFullShape(result.dtype, result.shape, result.ctx)[::-1]]
            elif isinstance(result.dtype, base.STuple):
                output_shapes = [base.subsFullShape(r, result.shape, result.ctx)[::-1] for r in result.dtype]
            else:
                raise ValueError("Invalid dtype")
            result.data = [torch.zeros(shape) for shape in output_shapes]
            for idx in output_indices:
                state = fn.getInit()
                for i in input_indices:
                    full_idx = idx + i
                    partial_data = [d[full_idx + (...,)] for d in input.data]
                    state = fn.apply(state, partial_data)
                for n, s in enumerate(state):
                    result.data[n][idx] = s
        return result
```

*Figure 22.* The definition of `Accum` primitive in the Python frontend.

