# OpenReview forum: "Adaptive Self-improvement LLM Agentic System for ML Library Development"
_ICML.cc/2025/Conference — ICML 2025 poster_

### Official Review · Reviewer_4wDa · 2025-03-09

**Overall Recommendation:** 4

**Summary:**

The authors propose an agentic system with adaptive self-improvement capabilities, specifically designed for synthesizing high-performance ML libraries. The proposed synthesis algorithm targets architecture-specific programming languages (ASPLs), with experiments conducted on Streaming Tensor Programs. The primary motivation is that domain-specific accelerators often change drastically with each new hardware generation, creating a pressing need for the rapid development of ML libraries in low-level specialized languages, often without access to large corpora of examples.

The paper presents an iterative approach that employs LLMs to generate new code solutions, filters out high-quality solutions, and then leverages these examples as demonstrations for increasingly complex tasks.

The proposed approach is evaluated on a suite of 26 tasks (curated by the authors from first principles), covering eight types of common operators (e.g., matrix multiplication, attention blocks, mixture-of-experts). The findings indicate that this method achieves higher pass rates (up to 96%) and a 3.9× improvement over a single-LLM baseline.


## Update After Rebuttal:
I find this paper very interesting and recommend its acceptance. Please make sure to incorporate the changes discussed during rebuttal in the final version of the paper.

**Claims And Evidence:**

Overall, the paper's main claim, i.e. that adaptive self-improvement leads to higher pass rates and higher code-correctness coverage, is well supported by pass@k results on a specialized but well-motivated benchmark. None the less, it is unclear how the system would scale on libraries that are much bigger or that require 50–100 times more operators.

**Essential References Not Discussed:**

Overall, the related works section is well-written and provides sufficient background on prior research related to self-improving LLM agents and their design for specialized code generation tasks. Some prior efforts, such as TVM and Spiral, and more recent papers on end-to-end auto-tuning code generators, may be relevant. Additionally, a broader set of reflection-based LLM coding pipelines, such as Reflexion, and Tree-of-Thought could be cited or compared.

**Experimental Designs Or Analyses:**

The experiments are conducted using 8 systematically created categories of ML operators, such as those involving shape manipulation and advanced arithmetic. Each category consists of multiple tasks that differ in certain details, such as whether partial streams are reused. The evaluation is quite thorough for a custom suite of 26 tasks.

**Methods And Evaluation Criteria:**

The primary metric for evaluating the proposed agentic system is functional correctness, measured by pass@k across 26 tasks. Pass@n is also reported, indicating how many tasks are eventually solved by at least one attempt. The authors further analyze the number of input tokens consumed per attempt and whether more complex examples yield better outcomes.

These criteria are well-suited for code-generation tasks, and the emphasis on pass@k aligns with standard practices in LLM-based coding research. However, the authors do not provide timing data or real hardware evaluations of the generated STeP code, leaving the claim of "high-performance" implementations unvalidated. The experimental design primarily assesses correctness rather than raw performance.

**Other Comments Or Suggestions:**

The paper could be strengthened by providing concrete results on the size or complexity of the final STeP programs, beyond just pass@k metrics. Metrics such as lines of code, the number of shape transformations, or specialized instructions would help quantify the difficulty.

**Other Strengths And Weaknesses:**

Strengths:
- The paper is well-written, provides sufficient background, and introduces the method in detail with appropriate examples. I enjoyed reading this paper.
- It presents novel ideas as well as interesting instantiations of well-studied techniques in a specialized domain. I particularly found the idea of a “guardian” agent for checking a global type-theoretic property to be a clever application of multi-agent prompting.

Weaknesses:
- The paper’s evaluation focuses almost entirely on the correctness of relatively small tasks. There is no direct measurement of the speed, efficiency, or memory overhead of the generated STeP libraries.
- The authors do not provide strong evidence of how the approach scales beyond these 26 tasks or how maintainable and comprehensible the generated solutions will be in practice.

**Questions For Authors:**

- Q1: Have you measured the actual run-time performance of any of these automatically generated kernels on real or simulated hardware?
- Q2: Do you anticipate any unique challenges in applying this approach to more mainstream, well-established languages like CUDA/HIP or to CPU vector intrinsics?
- Q3: Have you evaluated the readability or maintainability of the final code solutions?

**Relation To Broader Scientific Literature:**

The paper's approach relates to self-refining, multi-agent code generation systems, specifically in the context of LLM agentic methods for self-play and self-improvement. It provides sufficient citations and discussions of the broader scientific literature in this domain. The idea of using IR to represent partially structured code with a dedicated compiler reminds me of work using MLIR to improve the efficiency of Tensor Compiler Construction [1].

[1] VASILACHE, NICOLAS, et al. "Composable and Modular Code Generation in MLIR." arXiv preprint arXiv:2202.03293 (2022).

**Theoretical Claims:**

The authors do not provide formal mathematical proofs or new theorems.

---

> ### Author Rebuttal · Authors · 2025-03-28
>
> We thank Reviewer 4wDa for the positive comments and helpful feedback. We were encouraged that the reviewer enjoyed reading the paper, found our ideas novel, and took the time to review the appendix. We will include all the discussions and results below in the revised version.
>
> ## Run-time performance measurement
> > *”Q1: Have you measured the actual run-time performance of any of these automatically generated kernels on real or simulated hardware?”*
>
> We manually translated the generated implementation of Task 2 in Figure 16 of our paper to a simulator built on top of [DAM-RS](https://github.com/stanford-ppl/DAM-RS), which models the streaming behavior of each STeP primitive and assumes every operation and specialized function takes one cycle. This task implements *softmax(S)@V* where S and V are of shape [n,m] and [m], respectively; since S and V are streamed sequentially, the cycle count should scale with n*m. Simulation results match this expectation. Detailed result: https://anonymous.4open.science/r/ICML2025-rebuttal-4D6B/fig.png
>
> ## Generality of the framework
> > *”Q2: Do you anticipate any unique challenges in applying this approach to more mainstream, well-established languages like CUDA/HIP or to CPU vector intrinsics?”*
>
> Our approach has two parts: adaptive self-improvement learning and agentic system organization. The learning process is broadly applicable; the challenges lie in tailoring agentic systems to other languages.
>
> Mainstream languages like CUDA, HIP, and CPU vector intrinsics exhibit global properties such as arbitrary memory access, data layout sensitivity, and side effects. Similarly, STeP enforces a global affine type constraint. Our framework addressed this using a *guardian* agent that detects and corrects affine type violations. This concept generalizes: domain-specific guardian agents can monitor and enforce global properties of various languages, adapting the STeP solution more broadly.
>
> A second challenge is that LLMs may lean toward surface-level patterns in mainstream languages due to their existence in training data, potentially missing more optimal or novel transformations. As shown in Section 6.2, our structural IR can increase sample diversity and thus boost the LLM agentic system performance. Extending this, structural IRs and tailored code generators can guide LLMs toward more creative solutions beyond conventional patterns.
>
> ## Code maintainability and complexity
> > *”Q3: Have you evaluated the readability or maintainability of the final code solutions?”*
>
> > *”Metrics such as lines of code, the number of shape transformations, or specialized instructions would help quantify the difficulty.”*
>
> **Maintainability statistics**. We assessed code maintainability using two metrics: maintenance index without comments (MIwoc) and with comments (MI) [1]. The comment weight (MIcw) is defined as MI - MIwoc and falls in [0, 49); MI > 85 indicates good maintainability. Using all correct programs from our best model (self-improved agentic Claude Sonnet), we recorded the top MIwoc and MI per task. The mean MIwoc is 102, MI is 149, and MIcw is 47—indicating well-commented, maintainable code.
>
> **Complexity statistics**. We used the same set of programs as the maintainability statistics to measure complexity. We measured all three metrics the reviewer suggested:
> - Lines of code: Counted via primitive calls (excluding comments/blank lines)
> - Shape transformations: Counted by use of Promote, Repeat, RepeatRef, and Flatten primitives
> - Specialized instructions: Counted as the number of specialized functions in task descriptions
>
> Across all completed tasks:
> | Metric | Min | Max | Mean |
> |-----|----|----|----|
> | Lines of code | 4 | 17 | 8.67 |
> | Shape transformations | 0 | 6 | 1.13 |
> | Specialized instructions | 2 | 7 | 3.68 |
>
> Detailed result: https://anonymous.4open.science/r/ICML2025-rebuttal-4D6B/tab.md
>
> ## Scalability
> > *”…it is unclear how the system would scale on libraries that are much bigger or that require 50–100 times more operators.”*
>
> As discussed in the “Larger scale evaluation potential” section of our response to Reviewer qrAB, current LLMs still have the capacity to self-improve over hundreds more tasks. If the number exceeds the context length, better stratification and selection functions are needed to preserve experience quality within the context window limit.
>
> ## Related work
> > *”... TVM and Spiral, and more recent papers on end-to-end auto-tuning code generators, may be relevant. Additionally, a broader set of reflection-based LLM coding pipelines, such as Reflexion, and Tree-of-Thought could be cited or compared.”*
>
> We thank the reviewer for providing more relevant work. We will incorporate these papers into related work and give a more thorough discussion of how our work improves the results of them.
>
> ## Reference
> [1] https://www.verifysoft.com/en_maintainability.html

---

> > ### Comment · Reviewer_4wDa · 2025-04-08
> >
> > Thank you for your response to my questions. I don’t have any further questions at this time. After considering all of the discussions here, I’ve decided to keep my original score and recommend acceptance of this paper.

---

> > > ### Author Response · Authors · 2025-04-08
> > >
> > > We thank Reviewer 4wDa for the thorough review and recommendation for acceptance of our paper. The feedback and suggestions further improve this paper. We appreciate the reviewer's engagement with our work throughout this process.

---

### Official Review · Reviewer_6GnU · 2025-03-12

**Overall Recommendation:** 2

**Summary:**

The paper suggests an (agentic) system based on LLMs that self-improves using sampling to learn programming for (architecture) specific languages. It claims that this is a complex task for which little data is available therefore necessitating the need for a reasoning system.

## update after rebuttal

I acknowledge the effort the authors put into the response. However, I don't intend to update my score.

> While Claude Sonnet achieves 70% on our benchmark,

If the baseline is good on your benchmark, it is more or less easy. Analogies are not good arguments..

> ”Also the generality of the framework is unclear - is it only for that particular language?”

You need to evaluate on more tasks... Otherwise don't call it framework.

Generally, the idea of a rebuttal is not to fix the paper within a few days, e.g., adding essential experiments - there is intentionally no possibility to upload a revised paper version, which would be needed to properly assess major changes. It is more for clarifications or pointing out misunderstandings. Thus, do not expect that doing so will be seen as a fix to major issues in the paper that leads to a better score, though no doubt sooner or later you should do so.

Reviews and rebuttal read. Thank you. The paper has merit and if not ICML, it will still make its way. No update to score was done.

**Claims And Evidence:**

It is not clear, why this programming task should be so challenging (even for experienced programmers) as claimed in the intro - also in the light that Claude Sonnet achieves already 70%.
Also the generality of the framework are unclear - is it only for that particular language (judging from the evaluation it is, as there is just one dataset constructed toward that language).

**Essential References Not Discussed:**

---

**Experimental Designs Or Analyses:**

The comparison against other models is not fully clear. It appears that they are comparing against raw base models, e.g. GPT4o .This seems unfair as their agentic systems performs a lot of extra computation and has access to tools (like the verifier). Thus, while the improvement is still non-trivial, it is unclear, if a system fine-tuned say on samples that got filtered by the verifier or in some other way, would not outperform the proposed system.

**Methods And Evaluation Criteria:**

The benchmark is self-constructed and consists of just few tasks. This limits severely the generalizability.

**Other Comments Or Suggestions:**

None

**Other Strengths And Weaknesses:**

The paper should more clearly carve out early on what the contribution to the ML field are. It focuses too much on the domain-specific problem.

Minor comment: The claims like "we do human style learning with some ref" are too brief and vague but still appearing multiple times. If important discuss properly, otherwise maybe just mention it in the discussion or

**Questions For Authors:**

None - but see uncertainties above

**Relation To Broader Scientific Literature:**

Agentic AI is a hot topic.

**Theoretical Claims:**

no theory

---

> ### Author Rebuttal · Authors · 2025-03-28
>
> We thank Reviewer 6GnU for the constructive comments and helpful feedback. We are encouraged that the reviewer found our improvement non-trivial. Below, we respond to the raised concerns.
>
> ## Challenge of this programming task
> > *”It is not clear why this programming task should be so challenging.”*
>
> We appreciate the question. While Claude Sonnet achieves 70% on our benchmark, this does not imply that ML library development using ASPLs is easy. As a real-world example, as we pointed out in the paper, it took the community two years post-H100 release to optimize attention—a key LLM operator—to ~70% peak performance. As an analogy, although Claude Sonnet scores 81.7% Pass@1 on HumanEval-Mul (Table 6 in [1]), generating code from language instructions remains a hard problem. Our benchmark represents only a subset of the broader challenge: implementing transformer operators using a Python-embedded, side-effect-free ASPL. Many other ML operators and ASPLs involve more complex semantics. We plan to expand the benchmark with more difficult tasks to push system capabilities further.
>
> ## Generality of the framework
> > *”Also the generality of the framework is unclear - is it only for that particular language?”*
>
> We thank the reviewer for raising this important point. While our evaluation focuses on one language, STeP, we argue that the framework is general. As discussed in Section 2.1, we identify two essential features of ASPLs—primitives and specialized functions—and show in Section 2.3 how STeP embodies them. Due to space constraints, we refer the reviewer to the "Generality of the framework" section of our response to Reviewer 4wDa for the challenges and solutions of applying our framework to other languages.
>
> ## Comparison fairness
> > *”… This seems unfair as their agentic systems perform a lot of extra computation and have access to tools (like the verifier).”*
>
> As the reviewer pointed out, differences in tool access and computational load might have influenced the outcomes. Therefore, we conducted an experiment that aligned both aspects.
>
> For computation fairness, we matched the token count of the single model with the agent and self-improved models by resampling. All model variants (single, agent, self-improved) have access to the same verifier so it is fair; the difference lies in how the verifier is leveraged. Self-improved models incorporate it throughout the process, while others use it only at the end as a final judge. We chose Claude Sonnet and GPT-4o as base models. Below is the result:
>
> | Pass@n | Claude Sonnet Single | Claude Sonnet Agent & Self-improved | GPT-4o Single | GPT-4o Agent & Self-improved |
> |-----|----|----|----|----|
> | From | 0.73 | 0.73 | 0.23 | 0.23 |
> | To | 0.77 | **0.96** | 0.38 | **0.81** |
>
> Therefore, our agentic systems still perform better under this fair setting. Since we also agree aligned comparisons can provide a more comprehensive view, we will add these results in the revised version.
>
>
> ## Finetuning
> > *”…, it is unclear, if a system fine-tuned say on samples that got filtered by the verifier or in some other way, would not outperform the proposed system”*
>
> We conducted supervised finetuning (SFT) using GPT-4o and found it improved performance, but less than our self-improvement approach.
>
> Since we do not know the exact SFT algorithm of OpenAI service for FLOPs matching, we tried our best to favor the SFT method. We began with the same 133 correct samples from all completed tasks used in the first iteration of self-improvement. Different from self-improvement which only picks 1 correct program per completed task, we picked all 133 programs to form the training dataset of SFT. We created three SFT datasets with varying prompt compositions:
> - 133 (base prompt+question+answer)
> - 133 (question+answer)
> - 17 (question+answers deduplicated via AST)
>
> Each dataset was used to train a separate SFT model. After that, we sampled each model on all the uncompleted tasks. Below is the result:
>
> | Pass@n | Finetuned | Self-improved |
> |-----|----|----|
> | From | 0.35 | 0.35 |
> | To | 0.62 | **0.81** |
>
> We appreciate the reviewer’s suggestion and will include these results in the revised version.
>
>
> ## Paper organization
> > *”The paper should more clearly carve out early on what the contribution to the ML field are”*
>
> > *”The claims like "we do human style learning with some ref" are too brief and vague… “*
>
> We thank the reviewer for these helpful suggestions. In the revised version, we will emphasize our contributions to ML more clearly in the introduction and better define human-style learning in the discussion.
>
> ## Reference
>
> [1] Liu, A., Feng, B., Xue, B., Wang, B., Wu, B., Lu, C., Zhao, C., Deng, C., Zhang, C., Ruan, C. and Dai, D., 2024. Deepseek-v3 technical report. arXiv preprint arXiv:2412.19437.

---

### Official Review · Reviewer_hNb4 · 2025-03-14

**Overall Recommendation:** 3

**Summary:**

This paper proposes an adaptive self-improving agent system to unleash the ability of LLM to perform complex reasoning using limited data. It aims to automate the ML library development process using ASPL. To evaluate this, this paper builds a benchmark to conduct experiments to demonstrate the effectiveness of the proposed approach.

**Claims And Evidence:**

Yes, the claims made in the paper are supported by clear and convincing evidence. Especially, multiple clear flowcharts and algorithms demonstrate the operating principles and processes of the system.

**Essential References Not Discussed:**

No, the paper is well-cited and covers the essential references.

**Experimental Designs Or Analyses:**

I think the experiments in this paper have fully demonstrated the effectiveness of the various parts of the proposed system. Although this paper is about ML library development using an ASPL, are there other existing Agentic systems/workflows that can be applied to the current task?

**Methods And Evaluation Criteria:**

The proposed method is reasonable for the current problem. The benchmark simulates the library-chip co-design process, which is close to the real scenario and can verify the potential of the system if it is applied.

**Other Comments Or Suggestions:**

NA

**Other Strengths And Weaknesses:**

Since its main goal is to achieve complex reasoning with limited data, I think the proposed system should not be limited to the field of machine learning library design. Can some experiments be designed in the future to prove the reliability of the system in other fields and scenarios?

**Questions For Authors:**

I hope the authors can discuss the potential of the proposed system in other fields and scenarios.

**Relation To Broader Scientific Literature:**

The key contributions of the paper are related to self-improvement learning for LLMs and designing ML library using ASPLs.

**Theoretical Claims:**

This paper does not involve theoretical proof.

---

> ### Author Rebuttal · Authors · 2025-03-28
>
> We thank Reviewer hNb4 for the positive comments and helpful feedback. We are encouraged to hear the reviewer found our experiments and demonstrations clear and convincing. We also appreciate the reviewer’s careful review of the entire appendix content for prompt details.
>
> ## Other agentic methods for ML library development using ASPLs
> > *”…, are there other existing Agentic systems/workflows that can be applied to the current task?”*
>
> We thank the reviewer for raising this important point. As discussed in the paper, the tight timeline of the library-chip co-design process highlights the need for better automation. The community has begun to explore agentic solutions to this challenge in parallel [1, 2, 3]. Our adaptive self-improvement learning can enhance these efforts by making better use of correct samples. Additionally, existing systems often struggle with evolving language features, whereas our method, designed for new languages, adapts naturally to such changes.
>
> ## Other fields and scenarios potential
> > *”Can some experiments be designed in the future to prove the reliability of the system in other fields and scenarios?”*
>
> > *”I hope the authors can discuss the potential of the proposed system in other fields and scenarios.”*
>
> The proposed system can be extended to other scenarios that require complex reasoning with limited example data and well-defined evaluation metrics. We outline the general recipe below.
>
> As shown in Figure 1 of our paper, the agentic system organization is constructed in three main steps. First, system designers define the format of both the task and its expected output. Once the format is specified, the next step is to build a verifier for the task. With the format and verifier in place, designers can either use a single LLM or design LLM agents tailored to the domain—similar to how we handle the type constraints of the STeP language. After completing these three steps, the task can be handed over to our system, which will automatically carry out adaptive self-improvement learning.
>
> The adaptive self-improvement learning system also exposes several tunable hyperparameters which will be helpful when the results are not satisfactory. The most direct control is the number of parallel sampling. Users can also adjust the adaptive granularity parameter `m` for experience stratification. Additionally, domain-specific filtering heuristics—such as the minimal code length heuristic used by us—can be incorporated to further guide the learning process.
>
> We also conducted an experiment on the [AIME-2024 dataset](https://huggingface.co/datasets/Maxwell-Jia/AIME_2024) which contains 30 challenging problems from the American Invitational Mathematics Examination (AIME) 2024. We applied our adaptive self-improvement learning to the Claude Sonnet base model and increased Pass@n from **0.50** to **0.67**. This demonstrates the potential capabilities of our system on other tasks.
>
> Since we agree with the reviewer that demonstrating potentials in other domains can benefit the community, we will add the recipe and results in the revised version.
>
> ## References
>
> [1] https://developer.nvidia.com/blog/automating-gpu-kernel-generation-with-deepseek-r1-and-inference-time-scaling/
>
> [2] Lange, R.T., Prasad, A., Sun, Q., Faldor, M., Tang, Y. and Ha, D., 2025. The AI CUDA Engineer: Agentic CUDA Kernel Discovery, Optimization and Composition.
>
> [3] Ouyang, A., Guo, S., Arora, S., Zhang, A.L., Hu, W., Ré, C. and Mirhoseini, A., 2025. KernelBench: Can LLMs Write Efficient GPU Kernels?. arXiv preprint arXiv:2502.10517.

---

### Official Review · Reviewer_qrAB · 2025-03-25

**Overall Recommendation:** 4

**Summary:**

This paper introduces a novel task: utilizing LLM Agents that adaptively evolve to develop architecture-specific programming languages, addressing the challenges faced by human engineers in developing corresponding languages for rapidly evolving hardware. The experimental results appear very promising, and the proposed adaptive method is highly innovative, making this a noteworthy paper.

**Claims And Evidence:**

Yes.

All major claims of the paper, including adaptive self-improvement learning, curriculum-based example stratification, structured intermediate representation, and complex program discovery, are well supported by comprehensive experimental results with up to 3.9× improvement over baselines and 96% task completion rate. The authors provide detailed ablation studies and cross-model validations that demonstrate the effectiveness of their approach across different model architectures, with clear empirical evidence showing the superiority of hard-example training and the benefits of their structured intermediate representation design.

**Essential References Not Discussed:**

There aren't.

**Experimental Designs Or Analyses:**

Although the evaluation metrics are reasonable, the small size of the dataset needs to be noted, and experimenting with larger-scale datasets would likely further demonstrate the value of this paper's contributions.

**Methods And Evaluation Criteria:**

The Task is novel, so the paper establish a comprehensive benchmark consisting of 8 groups with 26 ML operator tasks for evaluation.
Althought it's a newly constructed dataset, the paper employs solid evaluation metrics and semantic diversity analysis, effectively demonstrating the system's capabilities and make the experimental validation convincing and meaningful.

**Other Comments Or Suggestions:**

Please see the weaknesses above

**Other Strengths And Weaknesses:**

Strengths:

1. The paper introduces a novel and significant task that addresses a critical need in developing ASPL for emerging hardware.
2. The proposed self-improvement methodology through adaptive curriculum learning and experience stratification is innovative and well-designed.
3. The experimental results demonstrate impressive performance improvements, achieving up to 3.9× enhancement over baselines and completing 96% of benchmark tasks.
4. The paper is well-written and clearly structured, effectively presenting complex concepts (such as those mentioned in background) and experimental validations.

Limitations:

1. The benchmark dataset, consisting of only 26 tasks across 8 groups, is relatively small and could benefit from a larger scale evaluation.
2. The paper could strengthen its literature review by incorporating more recent work on agent self-improvement, such as ADAS[1], AFLOW[2] to better position its contributions.
3. The inclusion of human programmer comparisons would provide valuable context and better demonstrate the practical significance of the system's achievements.

[1] Hu S, Lu C, Clune J. Automated design of agentic systems[J]. arXiv preprint arXiv:2408.08435, 2024.

[2] Zhang J, Xiang J, Yu Z, et al. Aflow: Automating agentic workflow generation[J]. arXiv preprint arXiv:2410.10762, 2024.

**Questions For Authors:**

Please see the weaknesses above

**Relation To Broader Scientific Literature:**

N/A

**Theoretical Claims:**

The paper does not include extensive theoretical analysis.

---

> ### Author Rebuttal · Authors · 2025-03-28
>
> We thank Reviewer qrAB for positive comments and helpful feedback on our work. We are encouraged to hear the reviewer found the task and method to be innovative and the experimental results to be promising.
>
> ## Larger scale evaluation potential
> > *”The benchmark dataset, consisting of only 26 tasks across 8 groups, is relatively small and could benefit from a larger scale evaluation.”*
>
> We thank the reviewer for highlighting the potential benefits of large-scale evaluation. We briefly addressed this point in Section 3.3 of the paper, and we will expand the discussion in the revised version. New tasks typically involve new ML operators and new hardware specialized functions. These can be incorporated into the existing task pool and handled using the same adaptive self-improvement learning process by selectively sampling only the new tasks. In our current experiments, the longest prompt is approximately 14k tokens, with each example averaging around 0.5k tokens. Given Claude Sonnet’s 200k-token context window, there is capacity to include hundreds of additional tasks.
>
> ## Related work
> > *”The paper could strengthen its literature review by incorporating more recent work on agent self-improvement, such as ADAS[1], AFLOW[2] to better position its contributions.”*
>
> We thank the reviewer for pointing out two relevant works—ADAS [1] and AFLOW [2]—that can enhance our literature review. The revised version will cite both papers.
>
> ## Human programmer comparisons
> > *”The inclusion of human programmer comparisons would provide valuable context and better demonstrate the practical significance of the system's achievements.”*
>
> We appreciate the reviewer’s suggestion. Our system completed each task in under **10 minutes** on average. In contrast, during our pilot study, a domain expert was unable to write a single program within **48 hours**, as they had to do trial-and-error and accumulate experience sequentially. Our system, by comparison, can perform these explorations in parallel. We agree with the reviewer that the comparison to human programmers offers valuable insight, and we will include these results in the revised version.
>
> In the future, we can also collaborate with HCI researchers to conduct more extensive experiments on the time and effort required by human programmers versus our system, aiming to better understand usability and cognitive load.
>
> ## References
>
> [1] Hu S, Lu C, Clune J. Automated design of agentic systems[J]. arXiv preprint arXiv:2408.08435, 2024.
>
> [2] Zhang J, Xiang J, Yu Z, et al. Aflow: Automating agentic workflow generation[J]. arXiv preprint arXiv:2410.10762, 2024.

---

> > ### Comment · Reviewer_qrAB · 2025-04-02
> >
> > The author's response has effectively addressed my potential concerns about this paper. Overall, this is an excellent paper, and I will maintain my score of 4 and recommend it for acceptance.

---

> > > ### Author Response · Authors · 2025-04-08
> > >
> > > We thank Reviewer qrAB for the thoughtful review and for recognizing the strengths of our work. We are grateful for the positive assessment and recommendation for acceptance. We appreciate the reviewer's time and feedback throughout the review process.

---

### Decision · Program_Chairs · 2025-05-01

**Decision:**

Accept (poster)

**Comment:**

This paper proposes an adaptive self-improvement agentic framework that leverages LLMs to automate the generation of ML libraries using architecture-specific programming languages (ASPLs), demonstrating strong results in synthesizing 25 out of 26 key LLM operators.

The strengths of the paper include:

- Novelty and relevance of the task: The paper introduces a timely and challenging problem of synthesizing ML libraries in ASPLs using LLMs, addressing an underexplored but impactful area (Reviewers qrAB, 4wDa).

- Innovative self-improvement framework: The adaptive curriculum-based learning loop, stratified experience sampling, and structured IR contribute meaningfully to agentic system design (Reviewers qrAB, hNb4).

- Strong empirical performance: The system achieves up to 3.9× improvements over baselines and 96% task completion, with detailed ablation studies and fair comparisons across model variants (Reviewers qrAB, 4wDa).

- Well-structured presentation and thoughtful design: The paper clearly articulates its design choices and motivations, making a compelling case for its adaptive methodology (Reviewers hNb4, 4wDa).

Before the rebuttal, common concerns raised by reviewers included the limited dataset size and benchmark scope (Reviewers qrAB, 6GnU, 4wDa), questions around generalizability beyond a single ASPL (STeP) (Reviewers 6GnU, hNb4), and lack of real hardware evaluation or human expert comparison (Reviewers qrAB, 4wDa). The authors effectively addressed these concerns in the rebuttal, offering clarifications, pilot simulation results, maintainability metrics, and concrete plans for broader evaluations.

Despite a mixed score from Reviewer 6GnU due to skepticism about benchmark difficulty and generality, the overall reviewer consensus supports the novelty, rigor, and potential impact of the work.

Therefore, AC recommend acceptance of this paper.